# SEMA: a Scalable and Efficient Mamba like Attention via Token Localization and Averaging

**Nhat Thanh Tran** [1]  **Fanghui Xue** [2]  **Shuai Zhang** [2]  **Jiancheng Lyu** [2]  **Yunling Zheng** [2]  **Yingyong Qi** [2]  **Jack Xin** [1]

## Abstract

Attention is the critical component of a transformer. Yet the quadratic computational complexity of vanilla full attention in the input size and the inability of its linear attention variant to focus have been challenges for computer vision tasks. We provide a mathematical definition of generalized attention and formulate both vanilla softmax attention and linear attention within the general framework. We prove that generalized attention disperses, that is, as the number of keys tends to infinity, the query assigns equal weights to all keys. Motivated by the dispersion property and recent development of Mamba form of attention, we design Scalable and Efficient Mamba like Attention (SEMA) which utilizes token localization to avoid dispersion and maintain focusing, complemented by theoretically consistent arithmetic averaging to capture global aspect of attention. We support our approach on Imagenet-1k where classification results show that SEMA is a scalable and effective alternative beyond linear attention, outperforming recent vision Mamba models on increasingly larger scales of images at similar model parameter sizes. Source code can be found at: https://github.com/nhatthanhtran/SEMA.

## 1. Introduction

Attention models of linear complexity in the input image size have been actively pursued in computer vision. One line of inquiry started with Swin (Liu et al., 2021) where window attention and shifting form a local-global approximation of vanilla attention (Vaswani et al., 2017) in the image domain. Another approach is the recursive attention of linear complexity, or a selective state space model known as Mamba (Gu & Dao, 2024), which has been applied to multi-directional scanning paths across an image for key-query computation (e.g. VMamba (Liu et al., 2024b)). Observing the similarity of Mamba and causal linear attention, (Han et al., 2024b) employed linear attention (Katharopoulos et al., 2020) in the macro-architecture of Mamba. Surprisingly, the resulting model MILA out-performed VMamba on $224^2$ images of ImageNet-1K and some downstream tasks. Since linear attention is known to lack expressive power or focusing capability ((Han et al., 2023) and references therein), it is mysterious that such a design works well. As the ablation study (Tab. 6 in (Han et al., 2024b)) showed, advanced and improved linear attention modules in standalone or other contexts (e.g. Flatten attention (Han et al., 2023)) did worse in the MILA environment.

In this paper, we introduce a generalized attention encompassing softmax and linear attention, and prove the generic dispersion phenomenon (i.e. equal attention to each token) as the number of tokens tends to infinity. To match this asymptotic behavior in the simplest manner, we adopt the arithmetic averaging as a global approximation, in conjunction with a localized (window) attention to maintain the desired focusing property of the vanilla softmax attention. We then place such a local-global approximation in Mamba like transformer and Swin architecture as MILA (Han et al., 2024b). Our design is thus explainable, besides being scalable and efficient as will be seen.

The key difference between our proposed method and others (Veličković et al., 2025; Ye et al., 2024; Han et al., 2023; 2024a) is that we utilize the theoretically confirmed dispersion property to approximate full attention. We prove that any global attention mechanism inevitably disperses.

Our main contributions in this paper are summarized below.

- We prove that for any continuous normalization function, the generalized attention disperses (i.e. pay equal attention to each token) in the limit of infinite range of tokens. For instance, both the vanilla softmax attention (Vaswani et al., 2017) and the commonly used linear attention (Katharopoulos et al., 2020; Li et al., 2020) disperse, among others.

[1]Department of Mathematics; University of California, Irvine; Irvine, CA 92697, USA [2]Qualcomm AI Research; San Diego, CA 92121, USA. Correspondence to: Nhat Thanh Tran <nhattt@uci.edu>.

*Proceedings of the $43^{rd}$ International Conference on Machine Learning*, Seoul, South Korea. PMLR 306, 2026. Copyright 2026 by the author(s).

- We use theoretical guidance from the long range token asymptotic limit to match dispersive behavior with arithmetic averaging for capturing the global aspect of attention layer in transformer models. Together with window attention to preserve the local aspect of attention on high frequency features, we put forth a novel scalable and efficient local-global attention (SEMA) in a Mamba type macro-structure (Gu et al., 2020; Han et al., 2024b).

- We demonstrate in ImageNet classification that SEMA is effective while image sizes scale up. It is flexible in finetuning on larger image input, improving accuracy and efficiency over recent scalable vision models (Liu et al., 2024b). SEMA is also competitive in downstream tasks.

## 2. Related Works

Transformer (Vaswani et al., 2017) with its softmax attention mechanism has been successful in language models. Its architecture is extended to applications in computer vision (Dosovitskiy et al., 2021; Liu et al., 2021). The ability of having direct access to all of the tokens via the attention mechanism is essential. However, there are a few challenging problems. The first is the quadratic computation complexity in terms of the input scale, while the second is the hidden ability in the attention to focus on the relevant information.

Many works resolve the quadratic complexity of vanilla attention (Vaswani et al., 2017) such as window attention (Liu et al., 2021; Beltagy et al., 2020; Dong et al., 2022), and linear attention (Katharopoulos et al., 2020; Li et al., 2020) simplifying the softmax into a separable form and applying the associative property of matrix multiplication. Linear attention has been improved by others such as Han et al. (2023; 2024b). Mamba (Liu et al., 2024b; Gu & Dao, 2024) is a linear complexity state space alternative to transformer. Inspired by Mamba, MILA (Han et al., 2024b) designed a linear attention block in a macro structure of the state space model. However, linear attention has the non-injectivity issue addressed in Han et al. (2024a).

The attention selection mechanism suffers greatly as the number of tokens increases. In very long context, the softmax attention is found unable to attend to important keys (Veličković et al., 2025; Ye et al., 2024). Lately Veličković et al. (2025) describes this phenomenon as dispersion. There are existing works attempting to resolve the dispersion problem observed in the softmax attention head as the context length increases. Veličković et al. (2025) proposed an adaptive temperature fix to increase the sharpness of softmax attention, whereas Ye et al. (2024) suggested to compute the attention as the difference of two attention matrices to

remove the noises in the attention matrices, and allow a query to attend to useful keys. The dispersion effect has been observed in linear attention in situations where softmax does not exhibit (See Figure 4 of Han et al. (2024a) and Figure 3 of Han et al. (2023)). In order to increase the focus of linear attention, Han et al. (2023) proposed to enhance the large attention coefficients while suppressing the smaller ones, while Han et al. (2024a) adopted subtraction instead of division for normalization of linear attention. Both works have convolution to help maintain focus.

## 3. Theoretical Analysis of Attention and Variants: Dispersion Phenomenon

We present a general formulation of the vanilla attention to capture many variants in a single framework. The goal is to allow theoretical results to be flexible when applied to the current popular variants as well as future designs. We keep the definition simple for ease of readability. We will be specific about certain choices when addressing the attention variants.

**Notation:** For any matrix $M \in \mathbb{R}^{n \times d}$ we denote $m_i$ is the $i$-th row of the matrix. For a vector $v \in \mathbb{R}^d$, then $v_i$ is the $i$-th element of the vector.

### 3.1. Generalized Attention and Dispersion

We present our main theoretical results, beginning with

**Definition 3.1.** Let $x \in \mathbb{R}^n$ and $\phi : \mathbb{R} \to \mathbb{R}^+$, we define $\Phi : \mathbb{R}^n \to [0,1]^n$ such that $\Phi(x) = \left[ \dfrac{\phi(x_1)}{\sum_{j=1}^n \phi(x_j)}, \dots, \dfrac{\phi(x_n)}{\sum_{j=1}^n \phi(x_j)} \right].$

**Notation:** Let $f, g : \mathbb{R} \to \mathbb{R}$, we say $f(x) = \Theta(g(x))$ iff there exists $c_1, c_2 \in \mathbb{R}$ and $x_0 \in \mathbb{R}$ such that $c_1 g(x) \le f(x) \le c_2 g(x)$ for all $x > x_0$.

For $e^{(n)} \in \mathbb{R}^n$, if $\Phi(e^{(n)})_k = \Theta(1/n)$, then we say $\Phi(e^{(n)})_k$ has the dispersion property. That is as $n$ increases the value $\Phi(e^n)_k$ goes to 0 for all $k$, i.e. $\Phi(\cdot) = 0$ as $n \to \infty$. We observe that when $\phi(x) = \exp(x)$, $\Phi(x)$ is the familiar softmax operator. If $e_k^{(n)} > 0$, then we can use $\phi : \mathbb{R}^+ \to \mathbb{R}^+$. We now present the definition of the generalized attention which covers many popular attention variants that we will discuss later in the paper.

**Definition 3.2.** ($\Phi$-normalized attention) Let $Q, K, V \in \mathbb{R}^{n \times d}$ be the query, key, value matrices. Let $\psi_q, \psi_k : \mathbb{R}^d \to \mathbb{R}^d$ ( or $(\mathbb{R}^d)^+$), $\phi : \mathbb{R}$ ( or $\mathbb{R}^+) \to \mathbb{R}^+$ be continuous. The

generalized $\Phi$-normalized attention is:

$$A_\Phi(Q,K,V) := \begin{bmatrix} \dfrac{\sum_{l=1}^n \phi(\psi_q(q_1)\psi_k(k_l)^T)v_l}{\sum_{j=1}^n \phi(\psi_q(q_1)\psi_k(k_j)^T)} \\ \vdots \\ \dfrac{\sum_{l=1}^n \phi(\psi_q(q_n)\psi_k(k_l)^T)v_l}{\sum_{j=1}^n \phi(\psi_q(q_n)\psi_k(k_j)^T)} \end{bmatrix}. \quad (1)$$

For the remainder of the paper, we will assume $\psi_q, \psi_k : \mathbb{R}^d \to \mathbb{R}^d$ ( or $(\mathbb{R}^d)^+$) are continuous functions. We observe that if $\psi_q, \psi_k$ are identity and $\phi(x) = \exp(x)$, then this is the vanilla softmax full attention. Similarly, if $\psi_q(x) = \psi_k(x) = ELU(x) + 1$, and $\phi(x) = x$, then this is linear attention. Here we measure similarity between the queries and keys using the natural choice of inner product, however we can extend this definition to any other continuous similarity functions (Luong et al., 2015). The following Theorem 3.3 still holds for a more general case. For clarity, we refrain from using a more general similarity function since the current attention mechanism uses inner product by default. The normalization factor $1/\sqrt{d}$ in vanilla attention is implicitly inherited in $\psi_q$ and $\psi_k$. We now prove the main theorem of the paper.

**Theorem 3.3.** *Let $\mathcal{X} \subset \mathbb{R}^d$ be a compact input feature space, and let $X^{(n)} \in \mathcal{X}^n$ be a matrix of input features for $n$ items. Let $e_{ij}^{(n)} = \psi_q(q_i)\psi_k((k_j^{(n)}))^T$ where $Q = \gamma(x_1^{(n)}, \ldots, x_n^{(n)})$ and $K^{(n)} = \kappa(x_1^{(n)}, \ldots, x_n^{(n)})$, where $\gamma : \mathcal{X}^n \to \mathbb{R}^{n \times d}$ and $\kappa : \mathcal{X}^n \to \mathbb{R}^{n \times d}$ are continuous functions, each expressible as a composition of $L$ layers $g_L \circ f_L \circ \cdots \circ g_1 \circ f_1$ where each layer contains a feedforward component $f_i(z_1, \ldots, z_n)_k = f_i(z_k)$ and a self $\Phi$-normalized attentional component $g_i(z_1, \ldots, z_n)_k = \sum_{1 \le l \le n} \alpha_{lk} v_i(z_l)$ where $\alpha_{lk} \in [0, 1]$ are $\Phi$-normalized attention coefficients and $v_i$ is a feedforward network. Then, for any $\epsilon > 0$, there exists an $n \in \mathbb{N}$ such that $\Phi(e_i^{(n)})_j < \epsilon$ for all $1 \le i, j \le n$. That is the $\Phi$-normalized attention coefficients must disperse in all transformer heads.*

We briefly outline the proof here with details left to the Appendix B. Since the functions in the theorem are continuous, they all preserve compactness. Thus $\phi$, continuous on a compact set, must attain its maximum and minimum. Therefore, $\Phi(e_i^{(n)})_j \le \frac{1}{n}\frac{\phi(b)}{\phi(a)}$, for some $a, b \in \mathbb{R}$. Hence the theorem holds for all $n > \frac{\phi(b)}{\phi(a)\epsilon}$.

*Remark* 3.4. The compactness assumption can be relaxed to boundedness since we can take the closure of the set to get compactness by the Heine-Borel theorem. The boundedness assumption is strict, because if $\mathcal{X}$ is unbounded, then Theorem 3.3 may not hold. For example if we let $\phi(x) = \exp(x)$, and $\psi_q, \psi_k$ as identity and $qk_j^T = \log(1/j^2)$, then

$$\frac{\phi(qk_j^T)}{\sum_{i=1}^n \phi(qk_i^T)} \ge \frac{\phi(qk_j^T)}{\sum_{i=1}^\infty \phi(qk_i^T)} = \frac{6}{\pi^2 j^2}. \quad (2)$$

Thus the query $q$ attends to the first key with at least $6/\pi^2$ probability.

For a neural network, $\phi$ is typically Lipschitz continuous. This implies that for any suitable choice of $\phi$, the attention will disperse. There are heuristics to resolve the dispersion problem, e.g. Ye et al. (2024) suggested a so called differential attention that computes the difference between two attention matrices. They argued that this allows the noise to cancel out. However, because differential transformer computes the difference between two attentions matrices, theoretically it is still $\Theta(1/n)$. Thus differential attention disperses as well. One advantage of differential attention over standard attention is that it allows the attention matrix to take negative values, thus alleviates the lower bound of standard attention which is $C/n$ for some $C \in \mathbb{R}^+$. The compactness assumption is realistic in many applications. For natural language processing, the dictionary is finite, thus the total number of tokens is finite, e.g. GPT2 uses a dictionary of 50257 byte-pair-encoding tokens (See et al., 2019). For vision, given a fixed image resolution, we represent each channel of a pixel with an integer from 0 to 255. Therefore, there are finitely many permutations of image representations.

The dispersion coefficient $\phi(b)/\phi(a)$ depends on the maximum and minimum values of function $\phi$ on the domain. In addition, if $\phi(x)$ is homogeneous, that is $\phi(\beta x) = \beta\phi(x)$ for all $\beta \in \mathbb{R}$, then the dispersion coefficient is independent of the scaling of the argument. Thus $\phi(x) = \exp(x)$ is one of the "optimal" choices, i.e. softmax attention. There is some empirical evidence that linear attention disperses much faster than softmax attention, e.g. Han et al. (2023) shows that linear attention lacks focusing ability on image tasks, whereas softmax attention does not display similar behavior, see the examples of Fig. 3 in Han et al. (2023). This is consistent with our analysis of dispersion coefficients. Both Han et al. (2023) and Veličković et al. (2025) addressed this problem by choosing a $\phi$ to amplify the maximum and suppress the minimum, thus increasing (decreasing) the dispersion coefficient (rate). The notations here agree with those in Veličković et al. (2025) whose Theorem 2.2 is a special case of Theorem 3.3. The dispersion rate also depends on the datasets and tasks. For example (Veličković et al., 2025) shows that on Gemma's Module file, $\phi(b)/\phi(a) \in [9.7, 2.6 \times 10^6]$ with mean and standard deviation $(2.95 \times 10^2 \pm 7.76)$. On ImageNet-1K test set, the pretrained MILA model exhibits a dispersion rate of $\phi(b)/\phi(a) \in [2.4 \times 10^3, 3.2 \times 10^6]$ with mean and standard deviation $(2.8 \times 10^5 \pm 3.49 \times 10^5)$. Moreover, across all attention heads, the dispersion rate is below $10^6$ and $2 \times 10^6$ with probabilities of 94.2 and 98.3 respectively. These suggest that dispersion effects may arise at much smaller token counts in language models than in vision models. However, for certain image problems such as medical imaging, the

number of tokens can reach to the order of $10^9$ (Wu & et al, 2024).

Lastly, we present an alternative perspective on the dispersion phenomenon from a probabilistic standpoint. Assuming that the un-normalized attention scores are drawn from certain distributions, we show that the $\Phi$-normalized attention disperses with high probability.

**Proposition 3.5.** *Let $X_1, X_2, \ldots$ be random variables (not necessarily i.i.d.) such that $X_i > 0$ with $\mathbb{E}[X_i] = \mu > 0$, $\sup_{i,j} \text{Cov}(X_i, X_j) < \infty$, and for every $n$, $|\{(i,j) : \text{Cov}(X_i, X_j) > 0 \text{ and } 1 \leq i < j \leq n\}| \leq n^\alpha$ for $1 < \alpha < 2$. For $0 < \epsilon \ll \mu$, with (high) probability $1 - \epsilon$, there exists $N_\epsilon \in \mathbb{N}$, such that for all $n > N_\epsilon$, we have the inequality:*

$$\frac{X_i}{n(\mu + \epsilon)} \leq \frac{X_i}{\sum_{j=1}^n X_j} \leq \frac{X_i}{n(\mu - \epsilon)}. \quad (3)$$

The proof leverages the weak law of large numbers to the sum of covariance-bounded random variables, by selecting $N$ with respecting to $\epsilon$ to derive the bound. Further details are provided in the Appendix B. The inequality with $n^\alpha$ on the right hand side essentially requires that the positive covariance entries be sparse. If $\phi(\psi_q(q^{(n)})\psi_k(k_j^{(n)})^T)$ are drawn from $X_j$ distributions under suitable covariance condition, it follows from Proposition 3.5 that the $\Phi$-normalized attention disperses with high probability.

In the following subsections, we overview recent variations of linear attention and state space models in the lens of dispersion before presenting our model in section 4.

### 3.2. Focused and Window Attention

Focused Attention (Han et al., 2023) is a modification of linear attention where $\psi_q(x) = \psi_k(x) = f_p(\text{ReLU}(x))$, with $f_p(x) = \frac{\|x\|}{\|x^{**p}\|} x^{**p}$ for some $p$, and $x^{**p}$ represents element-wise power $p$ of $x$. This still falls under Theorem 3.3 because $\psi_q, \psi_k$ are continuous. This is another ad hoc approach to reduce the effect of dispersion because for $p > 1$, $x^{**p}$ greatly increases large components while significantly decreases small components. Thus the dispersion coefficient (rate) gets larger (smaller). A particular choice of $p$ is 3. In addition, due to rank mismatch between linear attention and vanilla full attention matrices, Han et al. (2023) suggested to offset the attention matrix by a sparse depthwise convolution matrix $M_{DWC}$, that is

$$FA(Q, K, V) := A_\Phi(Q, K, V) + M_{DWC}(V). \quad (4)$$

Here $A_\Phi$ is computed using $\psi_q(x) = \psi_k(x) = f_p(\text{ReLU}(x))$, $\phi(x) = x$. As $n \to \infty$, the focused attention becomes a convolution layer as the attention disperses toward zero.

*Table 1.* Attention types and their corresponding dispersion bounds. Here $a$ and $b$ are the argmin and argmax on particular compact domain of $\phi$ respectively. $a, b$ may differ per method.

| Methods | $\phi(x)$ | Bounds |
|---------|-----------|--------|
| Softmax | $\exp(x)$ | $\frac{\phi(a)}{n\phi(b)} \leq \Phi(x)_k \leq \frac{\phi(b)}{n\phi(a)}$ |
| Linear | $x$ | $\frac{\phi(a)}{n\phi(b)} \leq \Phi(x)_k \leq \frac{\phi(b)}{n\phi(a)}$ |
| Focused | $x$ | $\frac{\phi(a)}{n\phi(b)} \leq \Phi(x)_k \leq \frac{\phi(b)}{n\phi(a)}$ |
| MILA | $x$ | $\frac{\phi(a)}{n\phi(b)} \leq \Phi(x)_k \leq \frac{\phi(b)}{n\phi(a)}$ |
| Differential | $\exp(x)$ | $\frac{\phi(a)^2 - \phi(b)^2}{n\phi(a)\phi(b)} \leq \Phi(x)_k \leq \frac{\phi(b)^2 - \phi(a)^2}{n\phi(a)\phi(b)}$ |

Swin (Dong et al., 2022) introduces window attention mechanism: each query only interacts with a few keys in a predefined window. For completeness, we formulate the window attention mechanism in this section, and clearly show that it does not disperse.

**Definition 3.6.** Let $Q, K, V \in \mathbb{R}^{n \times d}$ be the query, key, value matrices resp. Let $w \in \mathbb{N}$ be the window size such that $w$ divides $n$. Define the window attention as:

$$A_\Phi(Q, K, V, w) := \begin{bmatrix} \dfrac{\sum_{j \in J(1)} \phi(\psi_q(q_1)\psi_k(k_j)^T)v_j}{\sum_{i \in J(1)} \phi(\psi_q(q_1)\psi_k(k_i)^T)} \\ \vdots \\ \dfrac{\sum_{j \in J(n)} \phi(\psi_q(q_n)\psi_k(k_j)^T)v_j}{\sum_{i \in J(n)} \phi(\psi_q(q_n)\psi_k(k_i)^T)} \end{bmatrix}. \quad (5)$$

For some index set $J(m)$, for example $J(m) = \{Mw + 1, \ldots, (M+1)w\}$, where $M = \lfloor \frac{m-1}{w} \rfloor$.

Thus for the $i$-th row of attention, we have

$$A_{\Phi,i}(Q, K, V, w) = \sum_{j \in J(i)} \frac{\phi(\psi_q(q_i)\psi_k(k_j)^T)v_j}{\sum_{l \in J(i)} \phi(\psi_q(q_i)\psi_k(k_l)^T)}, \quad (6)$$

which is independent of $n$, thus there is no dispersion.

The formulation of window attention here can be quite general. Since the index set $J(m)$ can be constructed so that it creates different patterns such as sliding window or dilated sliding window (Beltagy et al., 2020). In particular, any attention mechanism limiting the query to only attend to a fixed number of keys falls under definition 3.6. This is because that after positional encoding, we can relabel the indices of keys so that the final attention computation does not change. A major drawback of window attention is that it is unable to reach global receptive field. Thus any method based on window attention needs other mechanisms to account for this weakness. We will address this in SEMA.

### 3.3. Mamba and Recursive Attention

For a complete derivation of state space model and Mamba, we refer the readers to Han et al. (2024b); Gu & Dao (2024).

Here we adopt the notations of (Han et al., 2024b). The state space model is a map from input $x(t) \in \mathbb{R}$ to output $y(t) \in \mathbb{R}$ through a hidden state $h(t) \in \mathbb{R}^{d \times 1}$ that can be written as the following system of equations:

$$h'(t) = Ah(t) + Bx(t), y(t) = Ch(t) + Dx(t), \quad (7)$$

where $A \in \mathbb{R}^{d \times d}, B, h(t), h't(t) \in \mathbb{R}^{d \times 1}, C \in \mathbb{R}^{1 \times d}, D \in \mathbb{R}$. After applying the zero-order hold discretization, and extending the map to high dimensional input $x \in \mathbb{R}^{n \times c}$, we obtain the following multi-dimensional system of discrete equations (Han et al., 2024b):

$$h_i = \tilde{A}_i \odot h_{i-1} + B_i(\Delta_i \odot x_i), \ y_i = C_i h_i / 1 + D \odot x_i, \ (8)$$

where $x_i, y_i, \Delta_i, D, \in \mathbb{R}^{1 \times c}, \tilde{A}_i, h_{i-1}, h_i \in \mathbb{R}^{d \times c}, B_i \in \mathbb{R}^{d \times 1}, C_i \in \mathbb{R}^{1 \times d}$, / denotes elementwise division, and $\odot$ the elementwise product. In contrast, the causal linear attention in recursive form is ($S_0 = 0, Z_i := \sum_{j=1}^{i} k_j^T$):

$$S_i = 1 \odot S_{i-1} + k_i^T(1 \odot v_i), y_i = q_i S_i / q_i Z_i + 0 \odot x_i, \ (9)$$

for given $Q, K, V \in \mathbb{R}^{n \times d}$, the queries, keys, and values respectively. This reveals the association between Eq. (8) and Eq. (9): $q_i \simeq C_i, k_i^T \simeq B_i$ and $v_i \simeq x_i$. It follows from Eq. (8) and $h_0 = 0$ (see Appendix B for details):

$$y_m = \sum_{i=1}^{m} q_m \tilde{k}_i^T \tilde{v}_i + D \odot v_m, \quad (10)$$

where $\tilde{k}_i^T \tilde{v}_i = \left( \prod_{j=1}^{m-i} \tilde{A}_{m-(j-1)} \right) \odot (B_i(\Delta_i \odot x_i))$, with $\prod$ as matrix product in the elementwise sense. The second term in (10) can be understood as a skip connection. The first term in (10) is the causal masked attention with $\phi(x) = x$ (without the normalization term). In Mamba, $\tilde{A}_i$'s are chosen so that every element falls between 0 and 1 (Han et al., 2024b), implying that the product of $\tilde{A}_i$ goes toward zero, i.e. the sum in equation (10) converges as $m \to \infty$. For fixed query ($q_m$), increasing the number of keys above $m$ does not change the value of the Mamba attention in equation (10). This implies that Mamba-like attention mechanism does not disperse. We want to point out that the actual implementation of linear attention in MILA (Han et al., 2024b) is not casual, thus it will disperse as shown in the next subsection. Equation (10) shows that there is exponential forgetting of previous keys in terms of Hadamard products of $\tilde{A}_i$. This is quite similar to **LagT** measure in HiPPO mechanism (Gu et al., 2020). Later, we shall use window attention to mimic such forgetting mechanism.

### 3.4. Mamba Inspired Linear Attention

Mathematically, MILA is

$$MILA(q_i, K, V) = \sum_{j=1}^{n} \frac{\psi_R(\psi_q(q_i))\psi_R(\psi_k(k_j))^T v_j}{\sum_{l=1}^{n} \psi_q(q_i)\psi_k(k_l)^T}$$
$$+ \psi_L(v_i), \quad (11)$$

where $\psi_R$ is RoPE (Su et al., 2024), $\psi_L$ is LePE (Dong et al., 2022), and $\psi_q(x) = \psi_k(x) = ELU(x) + 1$.

The MILA equation (11) resembles the focused-attention formulation in (4), with the only difference being the RoPE gating. We showed that focused attention disperses toward a convolution as the number of keys $n \to \infty$. Likewise, MILA also converges toward a convolution in the same limit. The argument proceeds as in Theorem 3.3. We provided a more details explanation in Appendix B.4.

We summarize attention mechanisms discussed here with their corresponding $\phi$ and theoretical upper and lower bounds of the $\Phi$-normalized attention coefficients in Tab.1.

## 4. SEMA Models via Token Localization and Averaging

In order to resolve dispersion in softmax full attention, Veličković et al. (2025) suggested an adaptive temperature as an ad hoc technique for improving sharpness. In our notation, their choice is $\phi(x) = \exp(x/\theta)$ for some optimal $\theta$. We observe that as $\theta \to 0$, the softmax attention converges to a hardmax, i.e. all of the attention is given to a single key, the remaining keys receive zero attention. The functional form of $\theta$ is chosen manually with each dataset. This limits the ability of the model to generalize on new datasets. Instead of resolving the dispersion property of attention, we fully embrace the property since we proved that any suitable choice of $\phi$ will cause dispersion.

We aim to design a model where attention computation is local and does not lead to dispersion. A way to achieve this is to limit the number of keys that a query attends to, i.e. window attention or some form of sparse attention, which resembles exponential forgetting in Mamba-like recursive attention. Since window attention has a narrow receptive field, it prevents the model from accessing global information. To alleviate this problem, one may adopt either a shifted window (Liu et al., 2021) or a mixing mechanism after each attention layer. Each such fix has its own weakness however. Shifted window requires manual design and multiple layers (Liu et al., 2021). Though a mixing mechanism recovers global receptive field to a large extent, the exact form is often unclear if one also aims at efficiency. We showed that full softmax attention will disperse, i.e. $\mathrm{softmax}(QK^T) = \Theta(1/n)$, where $n$ is the number of keys. Thus one can mimic the full softmax attention by simply averaging out all the tokens (as a low pass filtering): for a query (row vector) $q$, key matrix $K$, and value matrix $V$, approximate row-wise in the large $n$ regime as: **softmax**$(q K^T)V \approx \frac{1}{n} \sum_{j=1}^{n} v_j$. We call this averaging a *homogeneous mixing*, which reduces the computational complexity of full attention coefficients from $O(n^2 d)$ to $O(1)$, as the matrix product $Q K^T$ is down to a scalar factor

*Table 2.* Performance reported on standard $224^2$ image size input of ImageNet-1K.

| Method | Type | # Params | FLOPs | Top-1 (%) |
|---|---|---|---|---|
| ConvNeXt-T (Liu et al., 2022) | CNN | 29M | 4.5G | 82.1 |
| MambaOut-T (Yu & Wang, 2025) | CNN | 27M | 4.5G | 82.7 |
| VAN-B2 (Guo et al., 2023) | CNN | 27M | 5.0G | 82.8 |
| MixFormer-B4 (Chen et al., 2022) | CNN+Transformer | 35M | 3.6G | 83.0 |
| Swin-T (Liu et al., 2021) | Transformer | 29M | 4.5G | 81.3 |
| PVTv2-B2 (Wang et al., 2022) | Transformer | 25M | 4.0G | 82.0 |
| CSwin-T (Dong et al., 2022) | Transformer | 23M | 4.3G | 82.7 |
| Inline-CSwin-T (Han et al., 2024a) | Transformer | 21M | 4.3G | 83.2 |
| Flatten-CSwin-T (Han et al., 2023) | Transformer | 21M | 4.3G | 83.1 |
| NAT-T (Hassani et al., 2023) | Transformer | 28M | 4.3G | 83.2 |
| VMamba-T (Liu et al., 2024b) | Mamba | 31M | 4.9G | 82.6 |
| LocalVMamba-T (Huang et al., 2024) | Mamba | 26M | 5.7G | 82.7 |
| DefMamba-S (Liu et al., 2025) | Mamba | 32M | 4.8G | 83.5 |
| MILA-T (Han et al., 2024b) | Mamba-like | 25M | 4.2G | 83.3 |
| SEMA-T (Ours) | Mamba-like | 26M | 4.3G | **83.7** |
| ConvNeXt-S (Liu et al., 2022) | CNN | 50M | 8.7G | 83.1 |
| MambaOut-S (Yu & Wang, 2025) | CNN | 48M | 9.0G | 84.1 |
| PVTv2-B3 (Wang et al., 2022) | Transformer | 45M | 7.9G | 83.2 |
| CSwin-S (Dong et al., 2022) | Transformer | 35M | 6.9G | 83.6 |
| VMamba-S (Liu et al., 2024b) | Mamba | 50M | 8.7G | 83.6 |
| DefMamba-B (Liu et al., 2025) | Mamba | 51M | 8.5G | 84.2 |
| MILA-S (Han et al., 2024b) | Mamba-like | 43M | 7.3G | 84.4 |
| SEMA-S (Ours) | Mamba-like | 46M | 7.6G | **84.6** |
| ConvNeXt-B (Liu et al., 2022) | CNN | 89M | 15.4G | 83.8 |
| NAT-B (Hassani et al., 2023) | Transformer | 90M | 13.7G | 84.3 |
| VMamba-B (Liu et al., 2024b) | Mamba | 89M | 15.4G | 83.9 |
| MILA-B (Han et al., 2024b) | Mamba-like | 96M | 16.2G | 85.3 |
| SEMA-B (Ours) | Mamba-like | 102M | 16.9G | **85.4** |

*Table 3.* Model top-1 (%) comparison with and without finetuning on larger size input images from ImageNet-1K.

| Method | Type | Image Size | # Params | FLOPs | No tuning | Finetune |
|---|---|---|---|---|---|---|
| VMamba-T | Mamba | $384^2$ | 31M | 14G | 82.4 | 83.5 |
| SEMA (Ours) | Mamba-like | $384^2$ | 26M | 13G | **83.1** | **84.1** |
| VMamba-T | Mamba | $672^2$ | 31M | 43G | 77.9 | 83.6 |
| SEMA (Ours) | Mamba-like | $672^2$ | 26M | 40G | **78.9** | **84.1** |
| VMamba-T | Mamba | $768^2$ | 31M | 57G | 74.7 | 83.5 |
| SEMA (Ours) | Mamba-like | $768^2$ | 26M | 52G | **75.4** | **84.2** |
| VMamba-T | Mamba | $1024^2$ | 31M | 100G | 57.4 | 83.4 |
| SEMA-T (Ours) | Mamba-like | $1024^2$ | 26M | 93G | **64.7** | **83.8** |

*Table 4.* Model average inference time of 1000 runs on NVIDIA RTX A6000.

| Method | Type | Image Size | Runtime (ms) |
|---|---|---|---|
| MILA-T | Mamba-like | $224^2$ | 25.86 |
| SEMA-T (Ours) | Mamba-like | $224^2$ | 23.83 |
| MILA-T | Mamba-like | $512^2$ | 26.10 |
| SEMA-T (Ours) | Mamba-like | $512^2$ | 23.87 |
| MILA-T | Mamba-like | $1024^2$ | 60.73 |
| SEMA-T (Ours) | Mamba-like | $1024^2$ | 40.33 |
| MILA-T | Mamba-like | $2048^2$ | 244.35 |
| SEMA-T (Ours) | Mamba-like | $2048^2$ | 161.85 |

Concretely, for given $Q, K, V \in \mathbb{R}^{n \times d}$ and $w \in \mathbb{N}$, such that $w$ divides $n$, define:

$$SEMA(Q, K, V) := A_\Phi(Q, K, V, w) + \left[ \frac{1}{n} \sum_{j=1}^{n} v_j \right],$$
(12)

where $[\cdot]$ broadcasts the row $n$ times to permit matrix addition. In all our experiments, we use $\psi_q(x) = \psi_k(x) = x$, and $\phi(x) = \exp(x)$, or the window softmax attention. The first term in Eq. 12 processes information locally via window attention and the second term aggregates the global information to all of the tokens. We present an overview comparison of window attention and SEMA attention in Fig. 1. For the overall architecture of the model, we will utilize the design of MILA. We present the details architecture in Appendix F. We choose MILA because (1) the design uses

$1/n$, and $d$ the hidden dimension. The algorithms for computing window attention and such a homogeneously mixed window attention (referred to as SEMA attention from here on) are summarized in Algorithm 1 in Appendix G.

*Table 5.* Mask R-CNN 1x and 3x tasks on COCO dataset using input image of resolution ($1280 \times 800$). Transformer(T), Mamba(M), Mamba-like(ML), **bold best-2**.

| Method | Type | # Params | FLOPs | $AP^b$ | $AP^b_{50}$ | $AP^b_{75}$ | $AP^m$ | $AP^m_{50}$ | $AP^m_{75}$ | $AP^b$ | $AP^b_{50}$ | $AP^b_{75}$ | $AP^m$ | $AP^m_{50}$ | $AP^m_{75}$ |
|---|---|---|---|---|---|---|---|---|---|---|---|---|---|---|---|
| | | | | **(a) Mask R-CNN 1x** | | | | | | **(b) Mask R-CNN 3x** | | | | | |
| ConvNeXt-T | CNN | 48M | 262G | 44.2 | 66.6 | 48.3 | 40.1 | 63.3 | 42.8 | 46.2 | 67.9 | 50.8 | 41.7 | 65.0 | 44.9 |
| PVTv2-B2 | T | 45M | 309G | 45.3 | 67.1 | 49.6 | 41.2 | 64.2 | 44.4 | 47.8 | 69.7 | 53.0 | 43.1 | 66.8 | 46.7 |
| CSWin-T | T | 42M | 279G | 46.7 | 68.6 | 51.3 | 42.2 | 65.6 | 45.4 | **49.0** | 70.7 | **53.7** | 43.6 | 67.9 | 46.6 |
| LocVMamba-T | M | 45M | 291G | 46.7 | 68.7 | 50.8 | 42.2 | 65.7 | 45.5 | 48.7 | 70.1 | 53.0 | 43.4 | 67.0 | 46.4 |
| VMamba-T | M | 50M | 271G | **47.3** | 69.3 | **52.0** | **42.7** | **66.4** | **45.9** | 48.9 | 70.6 | 53.6 | 43.7 | 67.7 | **46.8** |
| MILA-T | ML | 44M | 255G | 46.8 | **69.5** | 51.5 | 42.1 | **66.4** | 45.0 | 48.8 | **71.0** | 53.6 | **43.8** | **68.0** | **46.8** |
| SEMA-T (Ours) | ML | 46M | 275G | **47.2** | 69.9 | 52.2 | 42.4 | 66.6 | 45.6 | **49.2** | **71.0** | **54.1** | **43.9** | **68.3** | **47.1** |

*Table 6.* Semantic Segmentation on ADE20K. SS and MS denote single-scale and multi-scale, resp. FLOPs are calculated with an input size of $512 \times 2048$, **bold best-2**.

| Backbone | Params | FLOPs | mIOU (SS) | mIOU (MS) |
|---|---|---|---|---|
| ResNet-50 | 67M | 953G | 42.1 | 42.8 |
| DeiT-S + MLN | 58M | 1217G | 43.8 | 45.1 |
| Swin-T | 60M | 945G | 44.5 | 45.8 |
| ConvNeXt-T | 60M | 939G | 46.0 | 46.7 |
| NAT-T | 58M | 934G | 47.1 | 48.4 |
| VMamba-T | 62M | 949G | **47.9** | **48.8** |
| SEMA-T | 56M | 956G | **48.2** | 48.5 |

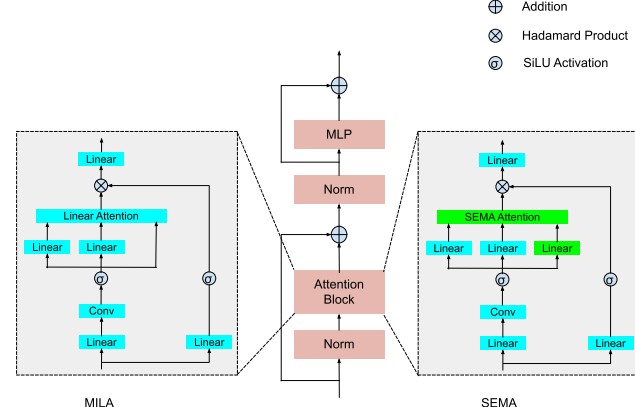

*Figure 2.* Mamba like transformers: MILA vs. SEMA, green blocks show our innovations.

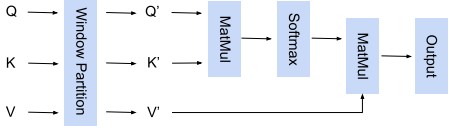

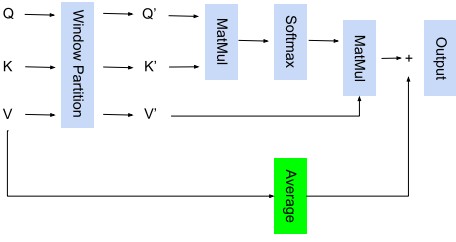

*Figure 1.* Homogeneous mixing (averaging) in SEMA attention (bottom) vs. window attention (top).

a simple linear attention that we will replace with SEMA attention; (2) our choice of window attention inspired by the forgetting property of Mamba naturally lands SEMA into the Mamba-like category. An overview of our model is in Fig. 2. The main difference lies on the Attention Block, where we used SEMA attention in lieu of Linear attention for MILA. We noted that MILA does not use a linear projection for the values $V$ (inspired by Mamba) while our formulation uses a linear projection for the values. For this reason, SEMA's parameter size is slightly higher than MILA's.

The latter part of the formulation in Eq. 12 is applied independently to each attention head. Consequently, a single

key representation is shared within each head. This design choice is closely related to KV cache mechanism. In particular, Multi-Query Attention (Shazeer, 2019) shares a single set of keys and values across all attention heads, thereby reducing memory and bandwidth requirements during inference. In contrast, Multi-Head Latent Attention (Liu et al., 2024a) stores keys and values in the KV cache using a low-rank latent representation, then reconstructs them via low-rank projection during attention computation. When the original keys are similar, this compression-projection (CP) process may produce nearly identical keys and values for attention, effectively behaving as an averaging operation. In this regime, the resulting attention computation is equivalent to the SEMA formulation. In general, the CP process resembles a sparsely weighted average which may be a future extension for SEMA.

## 5. Experiments

In this section, we will test SEMA on classical benchmarks in computer vision such as ImageNet-1K, COCO, and ADE20K. Due to limited resources available, we are unable to perform a thorough tuning of hyper-parameters and so have adopted those of MILA (Han et al., 2024b). We also report performance based on a single run in most

of the experiments, this is typical for this line of research under physical resource constraints (Liu et al., 2024b; Han et al., 2024b). Notably, we observed that the performance of SEMA-T on Imagenet-1K, as presented in Tab. 2, was consistent across several runs. [1]

**ImageNet-1K:** To evaluate SEMA on ImageNet-1K classification (Deng et al., 2009), we trained our models under the same settings as MILA (Han et al., 2024b) and Swin Transformer (Liu et al., 2021). The dataset consists of 1.28M training images and 50K validation images with a total of 1000 classes. We trained all of our models from scratch for 300 epochs using AdamW (Loshchilov & Hutter, 2019) optimizer with a cosine learning rate decay schedule of 20 warm-up epochs and weight decay of 0.05. The initial learning rate is set to $4 \times 10^{-3}$. We apply standard augmentation and regularization as RandAugment (Cubuk et al., 2020), Mixup (Zhang et al., 2018), CutMix (Yun et al., 2019), and random erasing (Zhong et al., 2020). For window attention, we used the standard window size of 7. We compared the results with current SOTAs. SEMA-T achieves a top-1 accuracy of 83.7, outperforming 83.3 of MILA-T (Han et al., 2024b), and DefMamba (Liu et al., 2025). Similarly SEMA-S/B achieves a top-1 accuracy of 84.6/85.4, surpasses 84.4/85.3 of MILA-S/B. We summarized the result in Tab. 2.

**ImageNet-1K on Larger Resolutions:** Classical tokenization of images, such as patching in Swin, converts every $4 \times 4$ pixels patch into a token via convolutions. Thus, increasing image resolution while maintaining the patch size increases the number of tokens. For example, under the standard settings, an image resolution of $224^2$ yields $n = 56^2$ tokens in stage one (Fig. 3 in the Appendix) where an image of size $1024^2$ generates $n = 256^2$ tokens. We evaluate models on higher image resolutions ($384^2, 672^2, 768^2, 1024^2$) of ImageNet-1K. Such experiments have been scarcely reported, except in VMamba (Liu et al., 2024b). We start with generalization under no-tuning. As increasing resolutions lower the performance, a prevalent phenomenon in existing networks (Liu et al., 2024b), we mainly compare the differences among the models. We observe that for image resolution of $384^2, 672^2$ and $768^2$, VMamba-T is about 1% behind SEMA-T. However, VMamba-T performs significantly worse than SEMA-T at $1024^2$. The gap is 7.3%. This reveals that VMamba-T struggles to process large images, while SEMA-T is much more resilient as image sizes scale up. We then finetune the models pre-trained on $224^2$ images with few epochs retraining on larger images (the corresponding size of the input in the generalization test), see Appendix D for details. The models all recover and surpass their respective performance levels at $224^2$. Table 3

shows that SEMA-T is 0.4-0.7% better than VMamba-T on all resolutions.

**COCO:** We also evaluate SEMA for the instance segmentation task on MSCOCO2017 (Lin et al., 2014) via the 1x and 3x Mask R-CNN training setting in Swin (Liu et al., 2021), with pretrained SEMA-T backbone on the ImageNet-1K. Due to resource limitations, we only conducted experiments using a couple of window size options. Our best results were obtained from window size of 12. We expect that finetuning the window size would yield better result. We compare SEMA-T to other SOTAs in Table 5. SEMA-T shows superiority in all measures for 3x task and stays in top-2 across all measures for 1x task. SEMA-T demonstrates performance that is comparable to or surpassing that of MILA-T across all metrics within Mamba-like architectures.

**ADE20K:** Lastly, we conduct semantic segmentation on ADE20K with a similar setup as Swin (Liu et al., 2021), see details in Appendix D. Table 6 shows that SEMA-T surpasses VMamba-T in the single scale regime with an mIOU of 48.2, while remaining competitive in the multi-scale regime with an mIOU of 48.5.

Overall, SEMA's performance on standard benchmarks such as ImageNet-1K, COCO, and ADE20K demonstrates two key findings. First, SEMA serves as a robust replacement for existing attention mechanism in the regime of relatively small number of tokens, i.e. $56^2$ for ImageNet-1K. In the large token regime, its theoretical guarantees provide an additional safety net. Second, as shown in Tab. 3, VMamba-T exhibits weaknesses when applied to large-resolution image inputs, whereas SEMA maintain significantly stronger adaptability and robustness.

We compare the runtime of SEMA against its counterpart MILA. In Tab. 4, we observe that at low resolution of $224^2, 512^2$, SEMA is about 10% faster than MILA with similar runtime. In the large resolution regime of $1024^2, 2048^2$ SEMA is about 33% faster than MILA, demonstrating the scalability of the proposed approach.

## 6. Ablation Study

In this section, we conduct experiments to show the benefit of modifications to the overall performance of the model. First, at the standard resolution of $224^2$, removing the value projection significantly reduces top-1 accuracy by about $-0.8\%$, while removing the averaging component has only a minor impact (about $-0.2\%$). This indicates that, in this regime, the performance gain primarily comes from the value projection (corresponding to the additional linear layer), while the averaging component-related to dispersion contributes little. Since the number of tokens in our network (see Fig. 3 in the Appendix) is small in the later stages, the typical window attention is nearly full attention, esp. stage

---

[1]All datasets were downloaded and evaluated by the university only.

4 has input resolution of $7 \times 7$ and the window size is 7, thus window and full attention are identical. This explains why we have a minor improvement under the input resolution of $224^2$. We demonstrated SEMA's stronger performance under larger input resolution in Tab. 3.

To further examine the role of the averaging (homogeneous mixing) component, we provide an additional ablation across different resolutions in Tab. 8. At $224^2$, the averaging term contributes only $+0.2\%$ top-1 accuracy. However, as the input resolution increases to $672^2$, its contribution becomes more pronounced: $+0.4\%$ after fine-tuning and $+2.2\%$ without fine-tuning.

These results are consistent with our theoretical analysis. When the input space $\mathcal{X}$ is fixed (Theorem 3.3), the dispersion coefficient is fixed, and the sequence length becomes the primary factor affecting dispersion. As the sequence length increases (e.g., higher resolution), dispersion becomes more significant, and the averaging term, serving as a global approximation, plays a larger role. Therefore, while the improvement at $224^2$ is not driven by dispersion, the gains at larger resolutions are increasingly attributable to the averaging component. After fine-tuning, we expect the input space $\mathcal{X}$ to shift, which can partially mitigate dispersion.

Although the benefit of homogeneous mixing in standard ImageNet-1K is marginal, (Dayag et al., 2026) shows that homogeneous mixing is highly effective when apply to medicals image segmentation tasks. In particular, as the image size increase the $F1$ score gain is much more significant. Under different dataset with various properties, we showed that adding a simple average help model capability. Thus demonstrate the robustness of the design. We include the results in Tab. 9 in Appendix C.

*Table 7.* Ablation study on images of $224^2$ resolution of ImageNet-1K.

| Method | Average | Value Projection | Top-1 |
|--------|---------|------------------|-------|
| SEMA-T | ✓ | | 82.9 |
| SEMA-T | | ✓ | 83.5 |
| SEMA-T | ✓ | ✓ | 83.7 |

*Table 8.* Ablation study of homogeneous mixing (averaging) on various images resolution of ImageNet-1K.

| Method | Average | $224^2$ | $672^2$ Finetune | $672^2$ No tune |
|--------|---------|---------|-------------------|------------------|
| SEMA-T | | 83.5 | 83.7 | 76.7 |
| SEMA-T | ✓ | 83.7 | 84.1 | 78.9 |

## 7. Conclusion

We formulated a generalized attention (GA) to subsume softmax and its linear attention variants, and proved that GA disperses, which inspires our SEMA design. SEMA uses window attention to avoid dispersion and averaging to capture global information. The simple averaging step is based on the large token asymptotic analysis of GA both deterministically and probabilistically, hence theoretically rooted. Experiments on ImageNet-1K, COCO, ADE20K showed the effectiveness, scalability and robustness of SEMA. In future work, we plan to (1) derive sharper bounds on dispersion coefficients, (2) extend averaging to learned and weighted averaging as well as sparsely weighted averaging in case of extremely long tokens, and (3) apply our framework to scientific data problems involving long tokens such as large size medical images (Wu & et al, 2024).

## Impact Statement

This paper presents theoretical and methodological advances for modeling long sequences in machine learning. The work does not involve new data collection. While the proposed techniques may be incorporated into systems used for a wide range of tasks, including medical image analysis, any societal or ethical impact will depend on the particular application, data, and deployment.

## Acknowledgments

The work was partially supported by NSF grants DMS-2151235, DMS-2219904, DMS-2309520, and a Qualcomm Gift Award. NTT was also funded by a Faculty Endowed Fellowship and the Graduate Scholar Success Fund from the University of California, Irvine.

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

## A. Limitations

We presented asymptotic result in the limit of number of keys $n \to \infty$. In reality, $n$ may be only moderately large but not enough to approach the theoretical limit. The theoretical bounds are not yet tight. Due to limited computational resources, we have yet to conduct extensive hyper-parameter tuning to fully optimize performance on large datasets (e.g. ImageNet). The scope of the current framework is being limited to vision tasks.

## B. Theoretical Analysis

### B.1. Softmax and Linear Attention

We review vanilla full attention mechanism (Vaswani et al., 2017) and its linear attention variant (Katharopoulos et al., 2020; Li et al., 2020) in the context of dispersion over infinitely long token range. We define these mechanism as follows.

**Definition B.1.** Let $Q, K, V \in \mathbb{R}^{n \times d}$ be the query, key, value matrices. The softmax full attention is:

$$
A(Q, K, V) := \begin{bmatrix} \dfrac{\sum_{i=1}^{n} \exp(q_1 k_i^T / \sqrt{d}) v_i}{\sum_{j=1}^{n} \exp(q_1 k_j^T / \sqrt{d})} \\ \vdots \\ \dfrac{\sum_{i=1}^{n} \exp(q_n k_i^T / \sqrt{d}) v_i}{\sum_{j=1}^{n} \exp(q_n k_j^T / \sqrt{d})} \end{bmatrix} = \mathrm{softmax}(QK^T / \sqrt{d}) V. \tag{13}
$$

**Definition B.2.** Let $Q, K, V \in \mathbb{R}^{n \times d}$ be the query, key, value matrices. The linear attention is:

$$
LA(Q, K, V) := \begin{bmatrix} \dfrac{\sum_{i=1}^{n} \psi_q(q_1) \psi_k(k_i)^T v_i}{\sum_{j=1}^{n} \psi_q(q_1) \psi_k(k_j)^T} \\ \vdots \\ \dfrac{\sum_{i=1}^{n} \psi_q(q_n) \psi_k(k_i)^T v_i}{\sum_{j=1}^{n} \psi_q(q_n) \psi_k(k_j)^T} \end{bmatrix}. \tag{14}
$$

for some suitable choice of $\psi_q, \psi_k : \mathbb{R}^d \to \mathbb{R}^{+d}$ such that the denominator is non-zero. A typical choice (Li et al., 2020; Han et al., 2024b) is $\psi_q(x) = \psi_k(x) = ELU(x) + 1$.

### B.2. Dispersion

In this subsection we will prove the theorem and proposition presented in the paper. We begin with

**Theorem B.3.** *Let $\mathcal{X} \subset \mathbb{R}^d$ be a compact input feature space, and let $X^{(n)} \in \mathcal{X}^n$ be a matrix of input features for $n$ items. Let $e_{ij}^{(n)} = \psi_q(q_i) \psi_k((k_j^{(n)}))^T$ where $Q = \gamma(x_1^{(n)}, \ldots, x_n^{(n)})$ and $K^{(n)} = \kappa(x_1^{(n)}, \ldots, x_n^{(n)})$, where $\gamma : \mathcal{X}^n \to \mathbb{R}^{n \times d}$ and $\kappa : \mathcal{X}^n \to \mathbb{R}^{n \times d}$ are continuous functions, each expressible as a composition of $L$ layers $g_L \circ f_L \circ \cdots \circ g_1 \circ f_1$ where each layer contains a feedforward component $f_i(z_1, \ldots, z_n)_k = f_i(z_k)$ and a self $\Phi$-normalized attentional component $g_i(z_1, \ldots, z_n)_k = \sum_{1 \le l \le n} \alpha_{lk} v_i(z_l)$ where $\alpha_{lk} \in [0, 1]$ are $\Phi$-normalized attention coefficients and $v_i$ is a feedforward network. Then, for any $\epsilon > 0$, there exists an $n \in \mathbb{N}$ such that $\Phi(e_i^{(n)})_j < \epsilon$ for all $1 \le i, j \le n$. That is the $\Phi$-normalized attention coefficients must disperse in all transformer heads.*

*Proof.* For a given head $(h)$ of a transformer block, since $\mathcal{X}$ is compact and all feedforward layers $f_i$ and $v_i$ are continuous, thus resulting spaces are also compact. Every self $\Phi$-normalized attention layer, $g_i$, computes a convex combination of the outputs of $v_i$ and if outputs of $v_i$ are on a compact space, the outputs of $g_i$ remain on the same compact space. Similarly, $\gamma$ and $\kappa$ are continuous, so they preserve compactness. The dot product of two vectors $\psi_q((q_i^{(n)})) \psi_k((k_j^{(n)}))^T$ coming from compact spaces must be compact as well. Thus, for all $1 \le i, j \le n$ the logits $e_{ij}^{(n)}$ must be bounded, that is there exists $M_h \in \mathbb{R}$ such that $\max_{i,j} |e_{ij}^{(n)}| < M_h$. Let $a, b \in [-M_h, M_h]$ such that $\phi(a) = \min_{x \in [-M_h, M_h]} \phi(x)$ and

$\phi(b) = \max_{x \in [-M_h, M_h]} \phi(x)$. We have for all $i, j$,

$$\frac{\phi(a)}{n\phi(b)} \leq \frac{\phi(e_{ij}^{(n)})}{\sum_{l=1}^n \phi(e_{il}^{(n)})} \leq \frac{\phi(b)}{n\phi(a)}. \tag{15}$$

Let $\epsilon > 0$, then there exists $N(h, \epsilon) \in \mathbb{N}$ such that for all $n > N(h, \epsilon)$, we have

$$\frac{\phi(e_{ij}^{(n)})}{\sum_{l=1}^n \phi(e_{il}^{(n)})} \leq \epsilon. \tag{16}$$

Let $N = \max_{1 \leq h \leq H} N(h, \epsilon)$ where $H$ is the total number of heads in the Transformer, thus for all $n > N$, we have

$$\frac{\phi(e_{ij}^{(n)})}{\sum_{l=1}^n \phi(e_{il}^{(n)})} \leq \epsilon. \tag{17}$$

This holds for all queries and keys in all of Transformer's heads. Hence we completed the proof. $\qquad\square$

Lastly, we will prove the

**Proposition B.4.** *Let $X_1, X_2, \ldots$ be random variables (not necessarily i.i.d.) such that $X_i > 0$ with $\mathbb{E}[X_i] = \mu > 0$, $\sup_{i,j} Cov(X_i, X_j) < \infty$, and for every $n$, $|\{(i, j) : Cov(X_i, X_j) > 0 \text{ and } 1 \leq i < j \leq n\}| \leq n^\alpha$ for $1 < \alpha < 2$. For $0 < \epsilon \ll \mu$, with (high) probability $1 - \epsilon$, there exists $N_\epsilon \in \mathbb{N}$, such that for all $n > N_\epsilon$, we have the inequality:*

$$\frac{X_i}{n(\mu + \epsilon)} \leq \frac{X_i}{\sum_{j=1}^n X_j} \leq \frac{X_i}{n(\mu - \epsilon)}. \tag{18}$$

*Proof.* Let $S_n := \sum_{i=1}^n X_i$ then we have

$$\text{Var}(S_n/n) = \frac{1}{n^2} \text{Var}(S_n) \tag{19}$$

$$= \frac{1}{n^2}\left( \sum_{i=1}^n \text{Var}(X_i) + 2 \sum_{1 \leq i < j \leq n} \text{Cov}(X_i, X_j) \right) \tag{20}$$

$$\leq \frac{Cn}{n^2} + \frac{2Cn^\alpha}{n^2} \leq 3C\frac{1}{n^{2-\alpha}}, \tag{21}$$

where $\sup_{i,j} \text{Cov}(X_i, X_j) \leq C$. For $0 < \epsilon \ll \mu$, and let $N = \lceil (\frac{3C}{\epsilon^3})^{\frac{1}{2-\alpha}} \rceil$ then by Chebyshev's inequality (Vershynin, 2026), we have

$$\mathbb{P}\left\{ \left| \frac{S_N}{N} - \mu \right| > \epsilon \right\} \leq \frac{\text{Var}(S_N/N)}{\epsilon^2} = \frac{3C}{N^{2-\alpha}\epsilon^2} \leq \epsilon. \tag{22}$$

Thus,

$$\mathbb{P}\left\{ \left| \frac{S_N}{N} - \mu \right| \leq \epsilon \right\} \geq 1 - \epsilon. \tag{23}$$

This implies with at least $1 - \epsilon$ probability $S_N/N > \mu - \epsilon$ and $S_N/N < \mu + \epsilon$, and so

$$\frac{X_i}{N(\mu + \epsilon)} < \frac{X_i}{S_N} < \frac{X_i}{N(\mu - \epsilon)}. \tag{24}$$

This holds for all $n > N$. Hence we completed the proof. $\qquad\square$

*Figure 3.* The 4 stage architecture of SEMA similar to Swin (Liu et al., 2021) and MILA (Han et al., 2024b).

### B.3. Mamba and Recursive Attention

In this subsection, we present the solution to a recursive state space model.

**Proposition B.5.** *Let $x_i, y_i, \Delta_i, D \in \mathbb{R}^{1 \times c}, A_i, h_i, h_{i-1} \in \mathbb{R}^{d \times c}, B_i \in \mathbb{R}^{d \times 1}, C_i \in \mathbb{R}^{1 \times d}$, for $i \in \{1, \ldots, m\}$, and*

$$h_i = A_i \odot h_{i-1} + B_i(\Delta_i \odot x_i), \quad y_i = C_i h_i + D \odot x_i \tag{25}$$

*where $h_0$ is given and $\odot$ denotes the elementwise (Hadamard) product. Then we have*

$$h_m = \left(\prod_{j=1}^{m} A_j\right) \odot h_0 + \sum_{i=1}^{m} \left(\left(\prod_{j=1}^{m-i} A_{m-(j-1)}\right) \odot (B_i(\Delta_i \odot x_i))\right) \tag{26}$$

*where $\prod$ is the elementwise multiple matrix product, under the convention that if the upper index is zero, the product acts as an identity. It follows that*

$$y_m = C_m \left(\prod_{j=1}^{m} A_j\right) \odot h_0 + C_m \left(\sum_{i=1}^{m} \left(\prod_{j=1}^{m-i} A_{m-(j-1)}\right) \odot (B_i(\Delta_i \odot x_i))\right) + D \odot x_m. \tag{27}$$

*Proof.* We prove by induction. First verify the base case,

$$h_1 = A_1 \odot h_0 + B_1(\Delta_1 \odot x_1). \tag{28}$$

Next for the inductive step, assuming

$$h_m = \left(\prod_{j=1}^{m} A_j\right) \odot h_0 + \sum_{i=1}^{m} \left(\left(\prod_{j=1}^{m-i} A_{m-(j-1)}\right) \odot (B_i(\Delta_i \odot x_i))\right), \tag{29}$$

we have

$$h_{m+1} = A_{m+1} \odot h_m + B_{m+1}(\Delta_{m+1} \odot x_{m+1}) \tag{30}$$

$$= A_{m+1} \odot \left(\left(\prod_{j=1}^{m} A_j\right) \odot h_0 + \sum_{i=1}^{m} \left(\left(\prod_{j=1}^{m-i} A_{m-(j-1)}\right) \odot (B_i(\Delta_i \odot x_i))\right)\right)$$

$$+ B_{m+1}(\Delta_{m+1} \odot x_{m+1}) \tag{31}$$

$$= \left(\prod_{j=1}^{m+1} A_j\right) \odot h_0 + \sum_{i=1}^{m+1} \left(\left(\prod_{j=1}^{m+1-i} A_{m+1-(j-1)}\right) \odot (B_i(\Delta_i \odot x_i))\right). \tag{32}$$

Substituting $h_m$ into the $y_m$ equation, we obtain

$$y_m = C_m \left(\prod_{j=1}^{m} \tilde{A}_j\right) \odot h_0 + C_m \left(\sum_{i=1}^{m} \left(\prod_{j=1}^{m-i} A_{m-(j-1)}\right) \odot (B_i(\Delta_i \odot x_i))\right) + D \odot x_m. \tag{33}$$

The proof is complete.

$\square$

## B.4. MILA

Given

$$MILA(q_i, K, V) = \sum_{j=1}^{n} \frac{\psi_R(\psi_q(q_i))\psi_R(\psi_k(k_j))^T v_j}{\sum_{l=1}^{n} \psi_q(q_i)\psi_k(k_l)^T} + \psi_L(v_i), \tag{34}$$

where $\psi_R$ is RoPE (Su et al., 2024), $\psi_L$ is LePE (Dong et al., 2022), and $\psi_q(x) = \psi_k(x) = ELU(x) + 1$.

We will provide a brief justification to show that MILA also disperses. Since we assume that the queries and keys come from some compact subset of $\mathbb{R}^d$, and since $\psi_R$, $\psi_q$, and $\psi_k$ are continuous functions, each of these mappings preserves compactness. The inner product also preserves compactness. Therefore the numerator $\psi_R(\psi_q(q_i))\,\psi_R(\psi_k(k_j))^T$ takes values in a compact subset of $\mathbb{R}$, and similarly the denominator $\psi_q(q_i)\,\psi_k(k_\ell)^T$ ranges over another (possibly different) compact subset of $\mathbb{R}^+$. The choice of $\psi_q, \psi_k$ guarantee that the denominators are positive. To ensure numerical stability, we add a small positive number to the denominator, e.g. $10^{-6}$. Finally, if we choose $\phi(x) = x$, and then the conclusion follows directly from Theorem B.3.

## C. Additional Homogeneous Mixing Experiments

In (Dayag et al., 2026), the authors integrate the SEMA block into the UNet framework to construct USEMA, where the attention blocks in the downsampling stages are replaced with SEMA blocks. The proposed model is evaluated on several medical image segmentation benchmarks, demonstrating the effectiveness of the homogeneous mixing component of SEMA across diverse datasets. For completeness, we include their reported comparisons in our paper as a reference. The evaluated datasets include Abdomen MRI MICCAI 2022 AMOS Challenge (Allan et al., 2019), which focuses on segmenting 13 abdominal organs from MRI scans; Endoscopy MICCAI 2017 EndoVis Challenge (Ma et al., 2024b), which focuses on segmenting seven different surgical instrument types; and Microscopy NeurIPS 2022 Cell Segmentation Challenge (Ma et al., 2024a), which focuses on cell segmentation in microscopy images. In Tab. 9, we observe that the $F1$ score improve significantly under SEMA with homogeneous mixing (average) setting.

| Dataset | Model | Average | DSC/F1 ($\uparrow$) | NSD ($\uparrow$) |
|---|---|---|---|---|
| Abdomen MRI | USEMA | ✓ | 0.7704 | 0.8345 |
| ($320 \times 320$) | USEMA | | 0.7574 | 0.8214 |
| Endoscopy | USEMA | ✓ | 0.6463 | 0.6621 |
| ($384 \times 640$) | USEMA | | 0.6218 | 0.6367 |
| Microscopy | USEMA | ✓ | 0.5791 | — |
| ($512 \times 512$) | USEMA | | 0.5443 | — |

*Table 9.* Ablation study of homogeneous mixing (averaging) for three medical dataset.

## D. Experiment Settings

**ADE20K** To evaluate semantic segmentation on ADE20K, we use UPerNet (Xiao et al., 2018) as segmentation framework in combination with MMSEG (Contributors, 2020) and follow the similar setting as Swin (Liu et al., 2021). We use a batch size of 8, and a window size of 14 for our model. We train using ADAM with learning rate of $6 \times 10^{-5}$, with $\beta \in [0.9, 0.999]$ and weight decay of 0.01.

**ImageNet-1K on Large Resolutions** When conducting no tuning or finetuning classification experiments on higher resolution ImageNet-1K, we follow the similar configurations as VMamba (Liu et al., 2024b). For no tuning, we load the pretrained VMamba models and conduct inference on $384^2, 672^2, 768^2, 1024^2$ using its validation scripts. For finetuning, the number of epochs is set to be 10 to ensure the training loss converges. During training, an AdamW optimizer is applied, with a learning rate of 0.001 and a weight decay of 0.05. We finetune the whole VMamba network, which is different from the linear finetuning in the original settings. Similarly, for SEMA we finetune for 30 epochs with a weight decay of $10^{-8}$ and a base learning rate of $10^{-5}$. We pick a window size of 8 for all of the resolutions since this is the closest divisors. The remaining finetuning strategy mirrors the approach used during the backbone training with a $224^2$ input resolution. We include our source code in the Supplemental materials.

| stage | output | SEMA-T | | SEMA-S | |
|---|---|---|---|---|---|
| 1 | $56 \times 56$ | stem, 64 | | stem, 64 | |
| | | dim 64
head 2
window size 7 | $\times 2$ | dim 64
head 2
window size 7 | $\times 3$ |
| 2 | $28 \times 28$ | downsampling, 128 | | downsampling, 128 | |
| | | dim 128
head 4
window size 7 | $\times 4$ | dim 128
head 4
window size 7 | $\times 6$ |
| 3 | $14 \times 14$ | downsampling, 256 | | downsampling, 256 | |
| | | dim 256
head 8
window size 7 | $\times 8$ | dim 256
head 8
window size 7 | $\times 21$ |
| 4 | $7 \times 7$ | downsampling, 512 | | downsampling, 512 | |
| | | dim 512
head 16
window size 7 | $\times 4$ | dim 512
head 16
window size 7 | $\times 6$ |

*Table 10.* Architecture of SEMA model.

## E. Computing Resource

We train and test all of the models on 8 NVIDIA RTX A6000 GPUs, each with 46G of memory. For standard image size of $224^2$ on ImageNet-1K, we use a batch size of 256 per GPU. For larger size image input $384^2, 672^2, 768^2, 1024^2$, the corresponding batch sizes are $128, 32, 32, 16$ per GPU respectively. For COCO dataset, we use an identical setup as in MILA (Han et al., 2024b). For ADE20K, we used similar setup as in Swin (Liu et al., 2021).

## F. Model Architecture

We present the architecture of our SEMA model in Fig. 3 and summarize the detailed structure in Tab. 10 which gives the parameter values in Fig. 3: $L_1 = 2, L_2 = 4, L_3 = 8, L_4 = 4$ for SEMA-T, among others. The 4-stage framework to build SEMA is similar to Swin (Liu et al., 2021) and MILA (Han et al., 2024b).

## G. Algorithm

---

**Algorithm 1** SEMA integrates window attention and homogeneous mixing (averaging), to be placed in a Mamba macro-setting so that long range global features are captured efficiently with local features.

---

**Input**: Input $x$
**Parameter**: window size $w$
**Output**: Attention

1: Compute $Q = \text{Linear}(x)$, $K = \text{Linear}(x)$, $V = \text{Linear}(x)$.
2: Divide $Q, K, V$ into $Q_w, K_w, V_w$ with window size $w$.
3: Apply RoPE to $Q_w, K_w$.
4: Compute window attention $WA = \text{softmax}(Q_w K_w^T)V_w$.
5: Compute $V_L = \text{LePE}(V)$.
6: Compute $V_A = \text{Average } V$ along the sequence dimension.
7: **Return** $WA + V_L + V_A$

---