# OpenReview forum: "SEMA: a Scalable and Efficient Mamba like Attention via Token Localization and Averaging"
_ICML.cc/2026/Conference — ICML 2026 regular_

### Official Review · Reviewer_gVwP · 2026-03-08

**Soundness:** 2
**Presentation:** 2
**Significance:** 3
**Originality:** 2
**Overall Recommendation:** 2
**Confidence:** 4

**Summary:**

This paper unifies common softmax attention and linear attention into a single paradigm. The authors argue that this paradigm leads to dispersion and propose SEMA to address this issue, validating it on the ImageNet-1K task.

**Compliance With Llm Reviewing Policy:**

Affirmed.

**Final Justification:**

According to the rebuttal, I will maintain my score.

**Key Questions For Authors:**

Do the conclusions in Sections 3.2, 3.3, and 3.4 indicate that generalized attention exhibits “dispersion”? If so, what does Proposition 3.5 aim to demonstrate?

**Limitations:**

Please refer to the “Strengths and Weaknesses” section. This will not be repeated here.

**Strengths And Weaknesses:**

Soundness:
I believe the problem addressed in this paper is not well aligned with the experimental setup. First, the paper claims to tackle the issue of “dispersion,” which arises when sequence lengths approach “infinite” lengths. However, the ImageNet-1K image classification task used in the experiments clearly does not sufficiently support this assumption.

Presentation:
I believe the wording in this paper lacks clarity and conciseness. For instance, what is the significance of Remark 3.4 and Proposition 3.5? Theorem 3.3 is unclear—is it intended to demonstrate that “dispersion” occurs in all generalized attention mechanisms? If so, what is the definition of “dispersion”?

Significance:
I find the research finding presented in this paper—that attention mechanisms encounter “problems” when processing ultra-long sequences—to be highly significant. However, I believe the proposed solution, “Token Localization and Averaging,” has limited significance and value. As mentioned earlier, the experimental setup using the ImageNet1-K classification task does not fully match the scenario of ultra-long sequences. Furthermore, the motivation behind Equation 12 and its (theoretical) effects are not sufficiently clarified in the paper (at least I did not fully grasp them through my reading). It appears that Equation 12 merely introduces a parallel branch to the attention mechanism, performing an averaging operation. Is a residual connection necessary in this case? Do both the averaging branch and the residual connection coexist?

Originality:
I find the originality of this paper to be average. Methodologically, I believe the novelty of SEMA attention is quite limited, representing an incremental improvement. Furthermore, I do not believe the paper offers profound or compelling insights beyond existing approaches such as window-based attention or MILA.

---

> ### Author Rebuttal · Authors · 2026-03-29
>
> Weakness:
>
> We thank you for recognizing the dispersions issues in attentions over long sequences. Our main contributions in the paper are not an incremental improvement of window attention and MILA, instead they are conceived based on the asymptotic analysis of attention. This new way of thinking goes beyond MILA, linear attention among other intuition based approaches. Specifically our novelties are:
>
> 1) Provide a general framework to unify many of existing attention designs such as softmax, linear, MILA. Under this
> framework, we derive a common property of attention mechanism (i.e. dispersion) to provide a theoretical perspective for numerous heuristic architecture modifications [1, 2, 3].
>
> 2) We showed that many existing choices of attention lead to dispersion. Most prior works treat dispersion as a problem to be corrected, we on the other hand use this property as an efficient global approximation.
>
> 3) We extracted the averaging step from asymptotic analysis in the long sequence limit, and arrived at an efficient and scalable global approximation, with additional wall-clock runtime to support the construction (in response to Reviewer gwPz).
>
> In short, we addressed the common weakness of many (approximate) attention designs to date.
> We proposed the first simple and efficient hybrid attention consistent with theoretical
> asymptotic behavior, thereby paving the way for future efforts on more accurate (higher order) asymptotically consistent low cost global attention.
>
> We address the remaining concerns below:
>
> 1) We provide brief definition of dispersion property in Section 3.1, right after Definition 3.1.
>
> 2) Equation 12 only describes the attention block computation. It consists of parallel branches of window attention and averaging (See Figure 1). Residual connections exist between each Attention Block and the MLP (See Figure 2).
>
> 3) The small size images (e.g. $224^2$) of ImageNet lie outside the regime of our theoretical guarantee. Despite this, our simple approximation via averaging maintains performance even though at a minor gain. For larger image sizes, we observe that SEMA performs well with much more gains (Table 3).  See also the table on inference times in response to gwPz and the table on segmentation results for medical images in response to reviewer VSHw.
>
> Key questions:
>
>
> Section 3.1 provides a general framework that captures many variations of attention. We proved in Theorem 3.3 that this generalized attention disperses. Section 3.2, 3.3, and 3.4 provide a summary of variants of attentions related to our work. We show that many of these heuristic designs are dispersive, e.g. Focused attention in Section 3.2. Section 3.3 on the other hand shows that Mamba does not exhibit the dispersion property and hightlights its connection to window attention. Thus, our use of window attention in Eq. 12 mimics the Mamba mechanism. Finally, section 3.4 demonstrates that MILA, with its design choices, also exhibits dispersion, which motivates the use of convolution to maintain local focus. Overall, we aim to provide examples in which these heuristics can be explained by the dispersion property, thereby offering explainability to many design choices in the field.
>
> Since Theorem 3.3 provides a deterministic theoretical result, the requirement of boundedness is strict. This is one of the reasons we provide Remark 3.4.
>
> Proposition 3.5 serves three purposes:
>
> 1) It allows us to remove the boundedness assumption in Theorem 3.3, with the trade off of probabilistic guarantee.
>
> 2) The results demonstrate that the order of $1/n$ is a good choice, thus leading to average.
>
> 3) A natural interpretation of Proposition 3.3 is as follows: given a query, keys are sampled from a distribution (e.g., words from a dictionary), thus the un-normalized score of $X_i = qk_i^T$ are random variables drawn from some distribution. These variables are not necessarily i.i.d. The covariance assumption allows for a partial correlation among keys, as expected in natural language (e.g. only a subset of words in a paragraph are strongly related).
>
> [1] Petar Velickovic, Christos Perivolaropoulos, Federico Barbero, and Razvan Pascanu. Softmax is not enough (for sharp size generalisation). ICML, 2025.
>
> [2] Dongchen Han, Xuran Pan, Yizeng Han, Shiji Song, and Gao Huang. Flatten transformer: Vision transformer using focused linear attention. ICCV, 2023.
>
> [3] Dongchen Han, Yifan Pu, Zhuofan Xia, Yizeng Han, Xuran Pan, Xiu Li, Jiwen Lu, Shiji Song, and Gao Huang. Bridging the divide: Reconsidering softmax and linear attention. NeurIPS, 2024.

---

> > ### Author Rebuttal · Reviewer_gVwP · 2026-04-03
> >
> > Thanks for the authors' reply. I will maintain my score.

---

### Official Review · Reviewer_X2Qc · 2026-03-11

**Soundness:** 2
**Presentation:** 4
**Significance:** 2
**Originality:** 2
**Overall Recommendation:** 4
**Confidence:** 3

**Summary:**

The paper introduces a generalized $\phi$-normalized attention framework that encompasses softmax and linear attention and proves that any such global attention inevitably “disperses” as the number of tokens grows, under mild continuity and boundedness assumptions. Motivated by this, the authors propose SEMA, a Mamba-like attention module that combines local window softmax attention (to preserve focusing) with a global channel attention based on MILA. SEMA achieves slightly better performance to their baseline, MILA, on ImageNet-1K and relatively large improvement on COCO detection. The consistent improvement demonstrates that a theoretically motivated local–global design is effective.

**Compliance With Llm Reviewing Policy:**

Affirmed.

**Final Justification:**

Thank you for the additional experiments and discussions. While there are still unclear aspects regarding the performance change from MILA to SEMA, and the validity of the proposed architecture is not yet fully understood, the observed improvement when adapting to high-resolution images without fine-tuning demonstrates the effectiveness of the proposed method. Considering these points, I change my rating to a weak accept.

**Key Questions For Authors:**

1.	The theory shows that any global $\phi$-normalized attention disperses as n becomes infinite, but in practice vision models operate at small n. Can the authors provide how much dispersion occurred in real trained models, in addition to the SEMA’s improved dispersion?
2.	In Table 6, what does “SEMA-T without averaging” indicate? Does the skip connection-like path remain while averaging operation is removed?
3.	More ablation studies are helpful to figure out which modifications (using softmax, adding linear layer, adding residual path, adding averaging in the residual path) are important.

**Limitations:**

yes

**Strengths And Weaknesses:**

# Strengths
1.	The paper provides interesting perspective of dispersion with a clean, general theoretical framework (Φ-normalized attention) that unifies softmax and linear attention.
2.	The local–global attention mechanism came up from theoretical insight is proved effective on ImageNet-1K, COCO and ADE20K dataset.
3.	The connection drawn between dispersion theory, Mamba-like architectures, and KV-cache–style averaging (including links to Multi-Query Attention and latent KV compression) is conceptually interesting and may influence how future long-context attention mechanisms are designed.

# Weaknesses
1.	The theoretical analysis is largely asymptotic in the number of tokens and provides only loose bounds. In recent LLMs, even with input sequences of thousand or ten thousands of tokens, the models can reproduce the input text almost exactly when asked to output it as is. This capability suggests that dispersion does not occur in such cases. Therefore, it is questionable whether dispersion actually occurs in vision tasks. However, it has neither been theoretically nor experimentally demonstrated that dispersion truly becomes a problem at the token lengths used in vision tasks.
2.	Alghthough the insight is interesting, the proposal in SEMA is simple; adding a channel attention-like path to MIRA. Also, the combination of local window attention and global channel attention is already proposed in DaViT [1] and FasterViT [2] introduce improved mechanism.
3.	The accuracy gain of SEMA against MIRA is marginal despite the additional computational cost.
4.	There is only one ablation study about w/ or w/o average pooling. The comparison to other anti-dispersion methods (e.g., adaptive temperature softmax, differential attention) is not conducted. More direct empirical comparisons in matched architectures would strengthen the case that SEMA is the preferable practical solution.

[1]Ding, Mingyu, et al. "Davit: Dual attention vision transformers." European conference on computer vision. Cham: Springer Nature Switzerland, 2022.
[2]Hatamizadeh, Ali, et al. "FasterViT: Fast Vision Transformers with Hierarchical Attention." The Twelfth International Conference on Learning Representations, 2024.

---

> ### Author Rebuttal · Authors · 2026-03-29
>
> We appreciate your feedback in recognizing the connection between our theoretical framework and KV cache style. We address your concerns below, and are happy to discuss any remaining questions.
>
> Weaknesses:
>
> 1. We agree that our main theorem is asymptotic. However, dispersion in practice is not binary but gradual. Both [1, 2] observe dispersion phenomenon in language models. In computer vision, [3, 4] observe that linear attention displays dispersion much more than softmax attention. This agrees with our theoretical analysis.
>
> 2. We clarify that our contribution is not only a structural combination, but a principled derivation. Prior works introduce local-global mixing heuristically. In contrast, SEMA derives global averaging as a first order approximation of $\Phi$ normalized attention. This provides 1) a unifying explanation for why such designs works, and 2) theoretically grounded alternative to ad hoc anti-dispersion fixes (e.g. temperature scaling, differential attention).
>
> 3. While the improvement at $224^2$ is modest, this regime is not where dispersion is expected to dominate. Importantly, SEMA shows clear advantages in the scaling regime. At $1024^2$ resolution, SEMA-T outperforms VMamba-T by $7.3$\% without finetuning revealing improved robustness as token count grows. In response to concerns raised by reviewers gWPz and VSHw, we provide additional empirical evidence. First, we include wall-lock runtime comparisons between SEMA-T and MILA-T, showing SEMA-T achieves approximately $33$\% speedup at high resolutions. Second, at $672^2$ resolution, SEMA-T outperforms its variant without average by $2.2$\% (without finetuning), futher supporting the effectiveness of the averaging mechanism in the large token regime.
>
> 4. Adaptive temperature softmax and differential attention were proposed for large language model. We are not aware of any such adaptations to vision. A method that tried to address dispersion in vision is Flatten Transformer [3]. The performance is much weaker than many baselines. We will update Table 2 to include this result.
>
> Key Questions:
>
> 1. Previous works [3, 4] showed that linear attention disperses much more than softmax on ImageNet-1K. This is consistent with our theoretical results. In addition, we reported dispersion coefficients for both LLM and CV in Section 3.1 (end of page 3).
>
> 2. SEMA-T without average uses window attention only. Everything else remains the same. Skip connections remain between each attention block and the MLP as shown in Figure 2.
>
> 3. We keep all of the design choices the same between the baseline model MILA and SEMA. The only difference is we replace linear attention of MILA with SEMA attention. Since we use a linear projection for values, we include an ablation study to show that this is beneficial. Adding a value linear layer increases the top 1 performance by 0.8\%.
>
> **Model top-1 (%) comparison with and without value linear layer on ImageNet-1K**
>
> | Method                               | Image Size | Top-1 (%) |
> |--------------------------------------|------------|-----------|
> | SEMA-T without value linear layer    | 224²       | 82.9      |
> | SEMA-T                               | 224²       | 83.7      |
>
> [1] Petar Velickovic, Christos Perivolaropoulos, Federico Barbero, and Razvan Pascanu. Softmax is not enough (for sharp size generalisation). ICML, 2025.
>
> [2] Tianzhu Ye, Li Dong, Yuqing Xia, Yutao Sun, Yi Zhu, Gao Huang, and
> Furu Wei. Differential transformer. arXiv:2410.05258, 2024.
>
> [3] Dongchen Han, Xuran Pan, Yizeng Han, Shiji Song, and Gao Huang. Flatten transformer: Vision transformer using focused linear attention. ICCV, 2023.
>
> [4] Dongchen Han, Yifan Pu, Zhuofan Xia, Yizeng Han, Xuran Pan, Xiu Li,
> Jiwen Lu, Shiji Song, and Gao Huang. Bridging the divide: Reconsidering
> softmax and linear attention. NeurIPS, 2024.

---

> > ### Author Rebuttal · Reviewer_X2Qc · 2026-04-03
> >
> > Thank you for the additional experiments and clarifications. Some of my concerns have been addressed. However, a few concerns still remain. So, I want to keep my score.
> >
> > First, regarding W1, I understand that dispersion occurs in linear attention even in visual recognition. However, the paper claims that dispersion arises in generalized attention (including both softmax and linear attention). As such, I feel that the claim is currently too broad and not fully aligned with the actual experimental evidence only from the visual recognition experiments.
> >
> > Moreover, if dispersion has already been identified as an issue in [3, 4], then I am not fully convinced about the novelty of revisiting this problem in this paper. In particular, considering that the formulation here only provides a loose bound, I think the paper would need to more clearly demonstrate how much the dispersion issue is actually mitigated in MILA in order to establish a stronger sense of technical progress.
> >
> > Regarding Q3, I believe SEMA introduces multiple modifications to MILA. For example, it changes global attention to window attention, replaces linear attention with softmax attention, adds extra linear layers, introduces an additional residual path, and introduces an averaging in the residual path. Since the performance gap between MILA and SEMA is not particularly large on ImageNet-1K (0.4 points (and 0.2 (I'm sorry. I found not 0.2 but 0.8 in the provided additional results.) points comes from the extra linear layers) for the tiny model and 0.1 points for the base model), it is difficult to assess whether the discussion around dispersion is truly responsible for the improvement unless the contribution of each of these design changes is clarified more explicitly.

---

> > > ### Author Response · Authors · 2026-04-08
> > >
> > > Prior works attempt to resolve dispersion at fixed resolution using heuristics, motivating us to generalize the attention framework to encapsulate many designs. Our contributions are threefold: (1) a unified formulation covering a broad family of attention mechanisms, (2) explicit analysis of dispersion as a function of sequence length, and (3) a design insight that global approximation (averaging) naturally stabilizes long sequence inputs. While analytically general to softmax and linear attention, our experiments focus on visual recognition.
> > >
> > > Unlike prior works [3,4], which identify dispersion or mitigate it heuristically at fixed scales, we formalize it in a unified framework and study its scaling with sequence length. This highlights a key insight: dispersion inherently induces a global averaging effect that can be leveraged.
> > >
> > > Rather than eliminating dispersion, we exploit it as a feature. Dispersion produces a global averaging effect, which we use as a stable approximation under long sequences, converting it into controlled averaging that preserves performance as sequence length grows.
> > >
> > > Our theoretical analysis provides a loose bound. While not tight, it captures the correct scaling trend with sequence length, the dominant factor driving dispersion, offering a simple, computationally inexpensive explanation of the observed behavior.
> > >
> > > Empirically, local window attention plus global averaging maintains performance at short sequences and remains robust as sequence length grows (see Table 3 and ablations below), supporting our claim that leveraging dispersion as global approximation is effective.
> > >
> > > Regarding Q3: We define SEMA attention as the sum of window attention and the average of the values in Equation 12. We show the macro structures of MILA and SEMA in Figure 2, where the differences are highlighted in green. The main differences are an additional value projection in SEMA and the replacement of linear attention in MILA with SEMA attention. There is no additional residual path; the averaging shown in the residual path (green line in Figure 1) reflects the SEMA attention defined in Equation 12. We use the same values for both the window attention and the averaging term. We believe this is the source of confusion, and we will clarify this more explicitly in the final version.
> > >
> > > To facilitate the discussion, we include ablation studies to isolate the contribution of each component. First, at the standard resolution of $224^2$, removing the value projection significantly reduces top-1 accuracy by about $-0.8\%$, while removing the averaging component has only a minor impact (about $-0.2\%$). This indicates that, in this regime, the performance gain primarily comes from the value projection (corresponding to the additional linear layer), while the averaging component-related to dispersion contributes little.
> > >
> > > To further examine the role of the averaging (homogeneous mixing) component, we provide an additional ablation across input resolutions. At $224^2$, the averaging term contributes only $+0.2\%$ top-1 accuracy. However, as the input resolution increases to $672^2$, its contribution becomes more pronounced: $+0.4\%$ after fine-tuning and $+2.2\%$ without fine-tuning.
> > >
> > > These results are consistent with our theoretical analysis. When the input space $\mathcal{X}$ is fixed (Theorem 3.3), the dispersion coefficient is fixed, and the sequence length becomes the primary factor affecting dispersion. As the sequence length increases (e.g., higher resolution), dispersion becomes more significant, and the averaging term, serving as a global approximation, plays a larger role. Therefore, while the improvement at $224^2$ is not driven by dispersion, the gains at larger resolutions are increasingly attributable to the averaging component. After fine-tuning, we expect the input space
> > > $\mathcal{X}$ to shift, which can partially mitigate dispersion.
> > >
> > > In regard to softmax vs. linear attention for window attention, in a short sequence softmax is computational more efficient.
> > >
> > > Overall, these results clarify that the averaging component (dispersion-related) has limited impact at standard resolution but becomes increasingly important as sequence length grows, which aligns with our theoretical findings.
> > >
> > > ### Table I: Ablation study of averaing on various image resolutions of ImageNet-1K
> > >
> > > | Method                     | 224²  | 672² Finetune | 672² No tune |
> > > |-----------------------------|-------|---------------|--------------|
> > > | SEMA-T without averaging    | 83.5  | 83.7          | 76.7         |
> > > | SEMA-T                      | 83.7  | 84.1          | 78.9         |
> > >
> > > ---
> > >
> > > ### Table II: Ablation study on images of 224² resolution of ImageNet-1K
> > >
> > > | Method    | Average | Value Projection | Top-1 |
> > > |-----------|---------|-----------------|-------|
> > > | SEMA-T    | ✔       | ✔               | 83.7  |
> > > | SEMA-T    |         | ✔               | 83.5  |
> > > | SEMA-T    | ✔       |                 | 82.9  |

---

### Official Review · Reviewer_VSHw · 2026-03-12

**Soundness:** 2
**Presentation:** 3
**Significance:** 3
**Originality:** 3
**Overall Recommendation:** 5
**Confidence:** 3

**Summary:**

This paper introduces SEMA (Scalable and Efficient Mamba-like Attention), a novel attention mechanism for computer vision that addresses two long-standing problems: the quadratic computational complexity of standard softmax attention and the poor focusing ability of linear attention alternatives. The authors begin by establishing a unified mathematical framework called generalized attentionn that encompasses both softmax and linear attention variants, then prove a fundamental theorem showing that all such attention mechanisms inevitably \meaning that as the number of tokens grows, each query ends up assigning equal weight to all keys, effectively losing its ability to focus on relevant information.

The key insight motivating SEMA is that rather than fighting this dispersion property, the authors lean into it. Since full attention asymptotically behaves like simple averaging in the large-token limit, they replace the expensive global attention computation with literal arithmetic averaging of all value tokens, which captures global context at negligible cost. This is combined with standard window (local) softmax attention to preserve fine-grained, focused feature extraction. The resulting SEMA attention is then plugged into the Mamba-like macro-architecture of MILA, yielding a model that is both theoretically grounded and computationally efficient.

On the empirical side, SEMA consistently outperforms comparable vision Mamba models across standard benchmarks including ImageNet-1K, COCO instance segmentation, and ADE20K semantic segmentation. Most notably, SEMA demonstrates significantly better scalability at higher image resolutions suggesting that its theoretical design advantages become especially pronounced as token counts grow large.

**Compliance With Llm Reviewing Policy:**

Affirmed.

**Final Justification:**

The authors have adequately addressed all of my concerns raised during the review. The 672² finetuning ablation directly validates the averaging component in the theoretically relevant regime, showing a meaningful 2.2% gap without finetuning and a persistent 0.4% improvement after finetuning. The additional medical imaging experiments across three diverse datasets (Abdomen MRI, Endoscopy, Microscopy) further demonstrate the robustness of the homogeneous mixing design, particularly at larger image resolutions. The high-pass/low-pass filter interpretation also provides a clean and intuitive characterization of the local-global trade-off. In light of these responses, I am raising my score accordingly.

**Key Questions For Authors:**

1. The ablation in Table 6 is conducted at 224² resolution, where the averaging term is effectively inactive due to window size matching feature map size at later stages. Can the authors provide an ablation at larger resolutions (e.g., 512² or 1024²) where dispersion is actually operative?

2. Simple averaging is justified as an approximation of fully dispersed attention. However, in tasks where the relevant signal is sparse — for instance, detecting small objects in dense scenes — averaging may suppress rather than aggregate useful information. Do the authors have any analysis or results that characterize when this approximation breaks down?

3. The theoretical framework is developed for vision tasks, but dispersion is also a known issue in long-context language modeling. Do the authors see any fundamental barriers to applying SEMA in that setting, and if not, are there any preliminary results?

**Limitations:**

No

The authors briefly mention future directions such as sharper dispersion bounds and weighted averaging extensions, but do not explicitly frame these as limitations of the current work. A dedicated discussion acknowledging the following would strengthen the paper: (1) the reliance on single-run evaluation and its implications for result reliability; (2) the conditions under which simple averaging is a poor approximation, particularly in sparse-signal settings; (3) the scope of the current framework being limited to vision tasks, with no discussion of whether the approach transfers to other domains.

**Strengths And Weaknesses:**

Strengths
Theorem 3.3 proves that attention disperses as token count grows, using standard tools from real analysis. The probabilistic extension in Proposition 3.5 adds further rigor, and the counterexample showing when the theorem fails reflects careful scholarship.
The core idea is novel. Most prior work treats dispersion as a problem to be corrected. This paper instead uses it to justify replacing global attention with simple averaging, turning a known limitation into a design principle. Unifying softmax, linear, focused, and differential attention under one framework also helps clarify how these methods relate to each other.
The scalability results support the approach. A gap of over seven percentage points against VMamba at 1024² resolution without fine-tuning is directly relevant to high-resolution vision applications.

Weaknesses
The ablation study is unconvincing. The averaging component contributes only 0.2% at standard resolution, which is too small to be conclusive given that results are based on a single run. At later network stages the window size matches the feature map size, so the averaging term contributes nothing there. The ablation should have been run at larger resolutions where the theoretical argument applies.
The empirical motivation for dispersion in Section 3 lacks details. Which attention heads were measured, at what point during training, and whether the reported ranges are representative are all unclear, making it hard to assess how well the observed dispersion motivates the design choices.
Simple averaging works when attention is fully dispersed, but it is a poor approximation when useful information is concentrated in a small number of tokens. How the method behaves in such settings is left unaddressed, which limits confidence in its broader applicability.

---

> ### Author Rebuttal · Authors · 2026-03-29
>
> We thank you for your appreciation of our theoretical works. We address your concerns below and are happy to address any remaining questions.
>
> Weaknesses:
>
> 1) The small image size of ImageNet is outside the regime of our theoretical guarantee, thus this shows that the crude approximation via averaging does not hurt performance. For larger image sizes, SEMA performs very well (Table 3).
>
> 2) The empirical results for dispersion in Section 3 were gathered across all attention heads in MILA-T. These results are from the ImageNet-1K test set during inference.
>
> 3) Averaging is a design choice that reflects the trade off between accuracy and efficiency. Since we aim for efficiency, we use this as an approximation. The theoretical results indicate that attending to a large number of tokens may harm the model, in that important tokens may not be represented in the final computation. Thus, an alternate design would construct a proxy to approximate the important tokens for computing attention, while using another mechanism to approximate global information. We would like to point out that Eq. 12 theoretically allows this flexibility, as element in the window can be chosen to be important tokens. Of course, identifying important tokens incurs additional computational overhead.
>
> Key Questions:
>
> 1) Due to time constraints of the rebuttal, we are unable to complete the full training at resolutions of $512^2$ or $1024^2$. However, we are able to complete the finetuning experiment on large image size of $672^2$. We use the baseline SEMA, with and without average, pretrained on $224^2$, and then finetune the models for 30 epochs on larger input of $672^2$. All training methodologies are the same as described in the finetune section of the paper (Table 3). We observe that without finetuning, the addition of averaging is very helpful, with a performance gap is about 2.2\%. After finetuning, the gap narrows; however, there remains an improvement of about 0.4\% for SEMA.
>
> **Model top-1 (%) comparison with and without finetuning on larger size input images from ImageNet-1K**
>
> | Method                    | Image Size | No tuning | Finetune |
> |---------------------------|------------|-----------|----------|
> | SEMA-T without average    | 672²       | 76.7%     | 83.7%    |
> | SEMA-T                    | 672²       | 78.9%     | 84.1%    |
>
> 2) From the design perspective, we use window attention as high pass filter, and average as a low pass filter. This balance renders the model versatile to different signal patterns.
>
> 3) We do not see any fundamental barriers to applying SEMA in LLM setting. We know that there are a few works that propose to help dispersion phenomenon (cited in the paper). There is a lack of research in vision regarding dispersion. So we focus mainly on extending SEMA to other vision tasks. We share some preliminary results on medical images that we found beneficial to your first question and also where SEMA applies robustly. To briefly summarize, we replace standard attention with SEMA in standard UNet structure for medical images. We observe that adding the average helps the model performance. In particular, as the image size increases, the gain of $F1$ score is much more significant. In addition under different dataset with various properties, we showed that adding a simple average helps model capability. Thus reveals the robustness of the design.
>
> Endoscopy : from MICCAI 2017 Endovis Challenge, segmentation of 7 different instrument types.
>
> Abdomen MRI : from MICCAI 2022 AMOS Challenge, segmentation of 13 abdominal organs from MRI scans.
>
> Microscopy: from NeurIPS 2022 Cell Segmentation Challenge, cell segmentation in microscopy images.
>
> **Ablation study of homogeneous mixing (averaging) for three medical dataset**
>
> | Dataset        | Image Size        | Model                      | DSC/F1 (↑) | NSD (↑) |
> |----------------|----------------|---------------------------|------------|---------|
> | Abdomen MRI    |(320 × 320) | SEMA                     | 0.7704     | 0.8345  |
> | Abdomen MRI   |(320 × 320) | SEMA w/o average term    | 0.7574     | 0.8214  |
> | Endoscopy      |(384 × 640) | SEMA                     | 0.6463     | 0.6621  |
> | Endoscopy      |(384 × 640) | SEMA w/o average term    | 0.6218     | 0.6367  |
> | Microscopy     |(512 × 512) | SEMA                     | 0.5791     | -       |
> | Microscopy     |(512 × 512) | SEMA w/o average term    | 0.5443     | -       |
>
> We provide a comparison of our model vs. other baselines below.
>
> **Results of cell segmentation on the Microscopy dataset**
>
> | Model         | F1 (↑) | # Params |
> |---------------|--------|----------|
> | SEMA | 0.5791 | 52M     |
> | U-Mamba Enc   | 0.5607 | 92M     |
> | Mamba UNet    | 0.5215 | 35M     |
> | Swin-UMamba   | 0.5186 | 60M     |
> | MLLA-UNet     | 0.4857 | 47M     |
> | UNETR         | 0.4357 | 88M     |
> | nnFormer      | 0.5332 | 60M     |
>
> Thank you for your suggestions, we will include those in the limitation section.

---

> > ### Author Rebuttal · Reviewer_VSHw · 2026-04-07
> >
> > I thank the authors for their responses. My concerns have been largely addressed.
> > The 672² ablation showing a 2.2% gap (without finetuning) and 0.4% (after finetuning) directly validates the averaging component in the regime where the theoretical argument actually applies — this should be included in the final paper.
> >
> > The medical imaging results across three datasets further support the robustness of the design, and the high-pass/low-pass filter framing is a helpful way to characterize the local-global trade-off.
> >
> > I am satisfied with the responses.

---

> > > ### Author Response · Authors · 2026-04-07
> > >
> > > We sincerely thank you for the follow-up, and
> > > are glad that our responses have adequately addressed the concerns.
> > > In light of this, could you please adjust the score ?
> > >
> > > We appreciate the constructive feedback throughout the review process. If
> > > there are any additional points we could address or further clarifications we can
> > > provide that would help strengthen the paper and improve your overall assessment, we would greatly appreciate your guidance. We will carefully incorporate
> > > all feedback and revisions in the final version.

---

### Official Review · Reviewer_gwPz · 2026-03-13

**Soundness:** 3
**Presentation:** 3
**Significance:** 3
**Originality:** 2
**Overall Recommendation:** 5
**Confidence:** 4

**Summary:**

This paper proposes SEMA, a Mamba-like attention mechanism for vision that combines localized window attention with a simple global averaging operation. The authors first introduce a generalized attention framework and argue, both theoretically and probabilistically, that many attention variants—including softmax and linear attention—tend to disperse as the number of tokens grows, which motivates replacing global attention with local attention plus homogeneous mixing. Based on this insight, they instantiate SEMA in a Mamba-like architecture and show on ImageNet-1K, COCO, and ADE20K that it achieves competitive or better accuracy than recent vision Mamba/Mamba-like baselines, while being especially robust when scaling to higher-resolution inputs.

**Compliance With Llm Reviewing Policy:**

Affirmed.

**Final Justification:**

Thanks for the rebuttal. I will increase my rating.

**Key Questions For Authors:**

See weaknesses

**Limitations:**

See weaknesses

**Strengths And Weaknesses:**

# Strengths

1. The paper is technically well-motivated, as it connects a theoretical analysis of generalized attention dispersion to a practical design choice: combining local window attention with global averaging. This theory-to-method pipeline is coherent, and the empirical results on ImageNet-1K, COCO, and ADE20K provide reasonably broad support that the proposed design is effective in standard vision settings.

2. The main novelty is not just a new module, but the perspective that global attention should be approximated through homogeneous mixing because generalized attention tends to disperse as token count grows. This theoretical framing gives the method a clearer rationale than many purely heuristic architectural modifications.

3. The paper is generally well organized, with a clear progression from generalized attention, to dispersion analysis, to the SEMA formulation, and finally to benchmark results. The high-resolution ImageNet experiments are especially valuable, since they highlight a practically relevant regime where the method appears more robust than VMamba-style alternatives.


# Weaknesses

1. While the theory is interesting, the practical gap between the asymptotic dispersion result and the final architecture is still somewhat indirect. The paper argues that averaging is theoretically consistent with the large-token limit, but this does not fully establish that simple averaging is the best or uniquely justified approximation in realistic finite-token vision regimes.

2. The empirical gains on standard-resolution ImageNet and the ablation are relatively modest, especially the direct gain from adding the averaging term (+0.2 top-1 in Table 6). In addition, the paper explicitly states that several experiments were run with limited tuning and mostly single runs, which weakens confidence in the robustness of the reported improvements.

3. Although the paper discusses related work, the distinction from closely related local-global or anti-dispersion designs could be sharpened further. In particular, the relationship between SEMA and prior window-attention, MILA, and other attention-fixing methods is not always delineated as crisply as it could be, which makes the exact novelty feel somewhat narrower than the paper sometimes suggests.

4. The paper does not provide throughput or latency comparisons, and instead mainly reports parameter count, FLOPs, and task accuracy. Since the method is positioned as a scalable and efficient Mamba-like attention design, the lack of wall-clock efficiency evaluation makes it difficult to assess its practical runtime benefits and deployment value, especially relative to closely related baselines.

---

> ### Author Rebuttal · Authors · 2026-03-29
>
> We thank you for recognizing that our method is mathematically driven. We address your concerns below and are happy to discuss further.
>
> 1. We aim to find an efficient and scalable global approximation for attention, thus average is a trade-off between efficiency and accuracy. This demonstrated in 4 below as the input size increases.
>
> 2. The small image size of ImageNet is not in the regime of our theoretical results. However, the proposed model is robust enough to handle such scenarios. We demonstrate in Tab. 3 that the image size increases, SEMA performance remains strong, especially in the case of zero-shot adaptation to an image size of $1024^2$ without tuning, compared to Mamba. We provide additional ablation study of adding the average term for medical imaging application in response to reviewer VSHw.
>
>
> 3. Our novel contribution lies in the generalized framework to incorporate wide range of attention design. This provides explainability of existing methods.
>
> 4. Thank you for the suggestion. We report  wall-clock runtime of SEMA in comparison to MILA, as we consider Mamba-like design. For image sizes of $224^2$ and $512^2$, SEMA is about $10$% faster than MILA. The runtime is similar at these scales due to overhead computation. However at scales $1024^2$ and $2048^2$, SEMA is about $33$% faster than MILA. We will update the final draft to include the new information.
>
> **Model average inference time of 1000 runs on NVIDIA RTX A6000**
>
> | Method          | Type        | Image Size | Runtime (ms) |
> |-----------------|------------|------------|--------------|
> | MILA-T          | Mamba-like | 224²       | 25.86        |
> | SEMA-T   | Mamba-like | 224²       | 23.83        |
> | MILA-T          | Mamba-like | 512²       | 26.10        |
> | SEMA-T    | Mamba-like | 512²       | 23.87        |
> | MILA-T          | Mamba-like | 1024²      | 60.73        |
> | SEMA-T    | Mamba-like | 1024²      | 40.33        |
> | MILA-T          | Mamba-like | 2048²      | 244.35       |
> | SEMA-T   | Mamba-like | 2048²      | 161.85       |

---

> > ### Author Rebuttal · Reviewer_gwPz · 2026-04-03
> >
> > Thanks for the rebuttal. I will increase my rating.

---

> > > ### Author Response · Authors · 2026-04-07
> > >
> > > We appreciate reviewer's feedbacks and suggestions. They help strengthen our paper.

---

### Decision · Program_Chairs · 2026-04-30

**Decision:**

Accept (regular)

**Comment:**

This paper presents a mathematically grounded approach to addressing the "dispersion" phenomenon in global attention mechanisms for long sequences. The authors provide a theoretical proof that all global continuous attention disperses as the number of tokens grows, motivating the design of SEMA, which is a hybrid module combining local window attention with a global global approximation (averaging). Reviewers initially raised concerns regarding the practical impact of dispersion at small token counts and the choice of benchmarks. However, the rebuttal successfully addressed these by providing additional experiments on high-resolution images and diverse medical imaging datasets, demonstrating robust performance improvements. The consensus is that the work offers strong theoretical insights and a highly effective, scalable architecture for computer vision tasks.